# Synovial Fluid Extracellular Vesicles from Patients with Severe Osteoarthritis Differentially Promote a Pro-Catabolic, Inflammatory Chondrocyte Phenotype

**DOI:** 10.3390/biom15060829

**Published:** 2025-06-06

**Authors:** Caitlin Ditchfield, Joshua Price, Edward T. Davis, Simon W. Jones

**Affiliations:** 1Department of Inflammation and Ageing, College of Medicine and Health, University of Birmingham, Birmingham B15 2TT, UK; c.ditchfield@bham.ac.uk (C.D.); j.price.4@bham.ac.uk (J.P.); 2NIHR Biomedical Research Centre, Birmingham B15 2TT, UK; 3The Royal Orthopaedic Hospital, Birmingham B31 2AP, UK; edward.davis@nhs.net

**Keywords:** osteoarthritis, extracellular vesicles, synovial fluid, articular chondrocytes, synovitis

## Abstract

Synovial inflammation is recognised as a pathological driver of osteoarthritis (OA), a degenerative joint disease involving cartilage degradation and joint pain. Since extracellular vesicles (EVs) have emerged as key mediators of cellular cross-talk, this study characterised synovial fluid EVs (SFEVs) in OA patients with varying disease severity and determined their functional effects on OA articular chondrocytes. Synovial fluid and articular cartilage were collected from patients undergoing knee surgery. SFEVs were isolated via ultracentrifugation and characterised by nanoparticle tracking analysis, ExoView, and Luminex analysis of protein cargo. Patients were stratified into mild/moderate- and severe-OA groups based on Oxford Knee Score and EQ5D. Chondrocytes were treated with SFEVs, and transcriptomic and secretome responses were analysed using RNA sequencing, Luminex, and ELISA. SFEVs from patients with severe OA were more abundant, smaller and exhibited increased tetraspanin expression. Synovial fluid and SFEVs induced distinct transcriptomic changes in chondrocytes. SFEVs from patients with severe OA promoted a pro-inflammatory and catabolic chondrocyte phenotype, with upregulation of *CRTAC1*, *COL6A3*, *TNC*, and *CXCL5*, greater secretion of IL-6, MMP1, MMP3 and MMP13, and pro-nociceptive mediators (NGF and Substance P). These findings suggest that SFEVs may contribute to OA progression by exacerbating cartilage damage and promoting pain sensitisation.

## 1. Introduction

Osteoarthritis (OA) is a degenerative joint disorder and a leading cause of disability worldwide [1], which poses a growing health burden due to ageing populations and rising obesity rates [2]. Unfortunately, current treatment options are inadequate, with no approved disease-modifying pharmacological drugs, and generic analgesics that exhibit limited efficacy and significant adverse side effects when taken chronically. Improved understanding of the molecular and cellular mechanisms that mediate OA is therefore a necessity so that more efficacious and targeted interventions can be developed to improve patients’ quality of life.

In attempting to develop new therapeutics, it is now recognised that OA is a complex heterogeneous condition involving multiple tissues of the joint, including articular cartilage degradation, remodelling of the subchondral bone, and synovial inflammation [3,4]. Emerging perspectives suggest that this heterogeneity is underpinned by distinct molecular endotypes [5,6], including “low repair”, “bone-cartilage”, “metabolic”, and “inflammatory” sub-types. Amongst these, the inflammatory sub-type, which is characterised by significant synovial fluid inflammation, is of particular interest due to the potential for synovial inflammation to exacerbate cartilage degeneration and to promote joint pain via the sensitisation of peripheral nociceptors [7]. Therefore, better understanding of the cross-cellular molecular signalling pathways that mediate and propagate the effects of synovial inflammation within the joint may provide the rationale for the development of novel targeted anti-inflammatory drugs in specific OA populations exhibiting an inflammatory endotype.

To this end, extracellular vesicles (EVs), which are small, lipid membrane-bound particles released by nearly all cell types into the extracellular environment, have emerged as novel candidate mediators of cellular cross-talk in OA. EVs are broadly heterogeneous and can be classified by both size and biogenesis. Small EVs (typically < 200 nm) and larger EVs (>200 nm) encompass populations such as exosomes (originating from endosomal multivesicular bodies) and microvesicles (formed by outward budding of the plasma membrane) [8]. In the context of OA, EVs have been implicated in disease pathogenesis, mediating cartilage degradation, inflammation, and nociception [9,10,11,12].

Asghar et al. (2024) [13] demonstrated that synovial fibroblast EVs contain distinct miRNA profiles, which contribute to chondrocyte damage. Furthermore, Cao et al. (2024) [10] revealed that EVs from the infrapatellar fat pad in OA patients impair cartilage metabolism and induce senescence. Distler et al. (2005) [14] showed that EVs contain OA-relevant proteases, including matrixmetalloproteases (MMPs) that promote cartilage degradation, and pro-inflammatory cytokines, including IL-1β and TNFα, which induce cartilage degeneration and are associated with peripheral pain sensitisation.

However, despite these insights, the characterisation of EVs that are present in OA synovial fluid and their role in mediating OA joint pathology remains poorly understood. This study had two main objectives. First, to comprehensively characterise synovial fluid extracellular vesicles (SFEVs) from patients with knee OA across different severity levels. Second, to investigate the functional effects of SFEVs derived from patients with severe or mild/moderate OA on the phenotype of articular chondrocytes, aiming to uncover novel mechanisms of SFEV-mediated communication within the OA joint.

## 2. Materials and Methods

### 2.1. Patient Recruitment and Sample Collection

Articular cartilage and synovial fluid were collected from OA patients undergoing total knee replacement surgery (NRES 17/SS/0456) at the Royal Orthopaedic Hospital, Birmingham, (United Kingdom) and Russell’s Hall Hospital, Dudley (United Kingdom). Synovial fluid was collected intra-operatively upon joint capsule opening, with volumes ranging from ~100 µL to 8 mL depending on fluid availability. Patients completed Oxford Knee Score (OKS), a visual analogue scale (VAS) of pain severity, and EQ5D questionnaires to capture OA joint severity and quality of life, and synovitis was scored by MRI across 11 compartments [15].

### 2.2. Isolation and Culture of Primary Human OA Chondrocytes

Cartilage was obtained from macroscopically intact regions of the joint. As anatomical compartmentalisation was not recorded, samples likely reflect a mixture of femoral, tibial, or patellar articular cartilage. Fresh cartilage was dissected into 1–3 mm^3^ pieces, and digested using 2 mg/mL collagenase (Merck, Gillingham, UK, C9891) in DMEM (Merck, Gillingham, UK D6429) for 7 h at 37 °C. Digested cartilage was filtered through a 70 µm cell strainer, the resultant filtrate was centrifuged at 400× *g* for 5 min, and then resuspended and cultured in chondrocyte growth media (DMEM, 10% FCS, 1% NEAA, 2 mM L-glutamine (ThermoFisher Scientific, Gloucester, UK, 25030024), 1% penicillin, and streptomycin (100 U/mL penicillin and 100 µg/mL streptomycin) (Merck, Gillingham, UK, P4333), 2.5 ug/mL amphotericin B (Merck, Gillingham, UK, A2942)). Growth media was replaced every 3–4 days and cells were passaged upon reaching 70% confluency. Chondrocytes were cultured in standard monolayer and used at early passage (P2) to minimise dedifferentiation. The same donor-derived chondrocyte population (BMI: 28.36) was used across all treatment conditions to minimise inter-donor variability. No data were collected on OKS, EQ5D, synovitis, or VAS for this donor to avoid introducing bias based on inflammatory status or clinical scores that could inadvertently influence downstream interpretations.

### 2.3. Synovial Fluid EV Isolation and Characterisation

Synovial fluid (*n* = 13) was mixed with 1 mg/mL hyaluronidase (Merck, Gillingham, UK, H3506) in PBS at 1:11, vortexed gently for 3 min, and incubated at 37 °C for 30 min. The hyaluronidase-treated synovial fluid was then centrifuged at 10,000× *g* for 10 min, and the supernatant was ultracentrifuged at 100,000× *g* for 16 h to pellet EVs [16], which were resuspended in PBS. For each patient, 1 mL of synovial fluid was used for EV isolation. Following ultracentrifugation, the EV pellet was resuspended in 100 μL of PBS, corresponding to 10% of the original volume.

Particles isolated by ultracentrifugation from the synovial fluid of 13 OA patients were analysed by nanoparticle tracking analysis (NTA), using the NSPro (Malvern Panalytical, Malvern, UK), to determine size distribution and concentration. Samples were diluted in PBS (Merck, D8537). The optimal dilution (1:200) was determined by NanoSight NS Xplorer software (Malvern Panalytical, Malvern, UK, https://www.malvernpanalytical.com/en/learn/events-and-training/webinars/w231010-explorenewnanosight (accessed on 2 June 2025)) utilising the software’s automated detection threshold and focus. Five videos capturing 400 tracks were taken of each sample at 25 °C with a syringe pump speed of 3 μL/min.

EVs were further characterised using the ExoView R100 reader (Unchained Labs, Malvern, UK) to analyse size, concentration, and CD9, CD63, and CD81 tetraspanin markers using the Leprechaun Exosome Human Plasma Kits (251-1045, Unchained Labs, Malvern, UK) at a dilution of 1:500, as previously described [17]. Chips were imaged using the ExoView R100 reader with ExoViewer 3.14 software and analysed using ExoView Analyser 3.0. Fluorescence gating was based on mouse IgG capture control, and sizing thresholds were set from a diameter of 50 to 200 nm.

As NTA detects particles based on size and Brownian motion, it may overestimate true EV concentration. Therefore, while we use the term ‘SFEVs’ to describe our preparations, these may contain a heterogeneous mix of particles. The presence of EVs was confirmed using tetraspanin profiling via ExoView and analysed using ExoView Analyser 3.0.

A subset of 11 patients were selected for SFEV characterisation. These consisted of 7 patients classified as patients with mild/moderate OA and 4 classified as patients with severe OA based on EQ5D and OKS. In some cases, individual patient-reported outcome measures were incomplete, resulting in minor discrepancies in *n*-values across different analyses (Table 1).

### 2.4. SFEV and Synovial Fluid Treatment of Chondrocytes

Chondrocytes were treated with either isolated SFEVs (5 × 10^9^ particles per well, 48 well plate) or with synovial fluid volumes predicted to contain an equivalent number of EVs, based on measured isolation efficiency. This corresponded to 4.5 μL per well for the severe-OA group and 8 μL per well for the mild/moderat-OA group. These volumes represented a small fraction of the total culture volume and were kept consistent within each group. Chondrocytes were treated with either pooled SFEVs or pooled synovial fluid from patients with mild/moderate or severe OA. Pooled samples were created by combining equal volumes from four classified donors per group (*n* = 4 mild/moderate, *n* = 4 severe). Treatments were carried out in serum-free media for 24 h to prevent interference from serum-derived EVs.

### 2.5. RNA Isolation and Bulk RNA Sequencing

Total RNA was extracted using a RNeasy Mini kit according to manufacturer’s instructions (Qiagen, Manchester, UK). RNA quantity and purity was determined by Nanodrop One (ThermoFisher, Altrincham, UK); all isolated RNA samples had A260/A280 ratios of 1.8–2.0. Library preparation and RNA-sequencing were performed by Beijing Genomics Institute (Hong Kong, China) using the DNBSEQ platform. Reads were mapped to the hg38 reference human genome using Bowtie. Differentially expressed genes (DEGs) were identified through DESEQ analysis and results analysed further by Ingenuity Pathway Analysis (IPA, Qiagen, Manchester, UK).

### 2.6. Luminex and ELISA

The concentrations of MMP1, MMP3, MMP13, BDNF, IL-6, and NGF proteins in conditioned media, synovial fluid or SFEV lysates were quantified using a customised Luminex assay (Bio-Techne, Centennial, CO, USA), whilst TIMP3 and ICAM1 were quantified by ELISA (TIMP3: A717-96, antibodies.com; ICAM1: A1716-96, antibodies.com). Prior to protein quantification, hyaluronidase-treated synovial fluid samples and isolated SFEVs were lysed with 0.5% TritonX and diluted 1:2 with dilution buffer. SFEV lysates were prepared from EVs isolated from the 13 patient samples used for EV characterisation. These included 7 classified as those with mild/moderate OA, 4 as those with severe OA, and 2 who did not meet the criteria for either group. For group-based comparisons, only lysates from the 11 classified patients were included in the analysis

### 2.7. Statistical Analysis

Statistical analysis was performed using GraphPad Prism v10, with *p*-values of <0.05 considered to be statistically significant, and FDR < 0.05 considered significant when analysing RNA sequencing data. Where appropriate, linear regressions were performed to test associations between variables. Two-way ANOVA was performed on ExoView data with Bonferroni post hoc tests to determine statistical significance between severe OA and mild/moderate OA tetraspanin expression. One-way ANOVA with Tukey’s multiple comparison post hoc tests were performed on RNA sequencing count data and Luminex/ELISA data.

## 3. Results

### 3.1. Patient Characteristics

Patients were stratified into mild/moderate- and severe-OA groups based on thresholds of OKS ≤ 21 and EQ5D ≥ 9 (severe), or OKS ≥ 22 and EQ5D ≤ 8 (mild/moderate), selected to create distinct groups based on the data distribution and allow a buffer between classifications. The subsequent experimental workflow is illustrated in Appendix A. Of 71 patients involved in this analysis, 29 were classified as patients with severe OA, 24 were classified as mild/moderate OA, and 18 did not meet the criteria for either group. An inverse relationship between OKS and EQ5D was observed (*p* < 0.0001, r^2^ = 0.3287), supporting the distinction between these two groups based on OA severity (Figure 1A).

### 3.2. Characterisation of Synovial Fluid Extracellular Vesicles

Patient characteristics of samples used for NTA and Exoview characterisation are summarised in Table 1. EVs were isolated by ultracentrifugation from the synovial fluid of *n* = 13 patients with mild/moderate OA and characterised by NTA (Figure 1B). NTA confirmed the successful isolation of SFEVs, with concentrations between 2.6 × 10^10^ to 1.7 × 10^11^ particles/mL, a mean size of 172 nm and a mean D50 of 136 nm (Figure 1C). Notably, NTA revealed distinct differences in the gross characteristics of SFEVs between severe- and mild/moderate-OA groups. A significant negative correlation was observed between EQ5D score and SFEV concentration (*p* = 0.0067, r^2^ = 0.5025, Figure 1D), indicating that lower quality of life was associated with lower SFEV concentrations. Conversely, a positive correlation was found between EQ5D score and SFEV size (*p* = 0.0137, r^2^ = 0.4383, Figure 1E) suggesting that patients with worse quality of life tended to have larger SFEVs. No significant correlations were detected between OKS or VAS scores and the SFEV parameters (Figure 1D and Figure 2E). Next, we used ExoView to characterise the surface expression of a panel of tetraspanin markers (*n* = 11) (Figure 1B). ExoView analysis of CD9, CD63, and CD81 tetraspanins revealed a significant effect of OA severity on the number of particles positively staining for tetraspanin expression, with elevated tetraspanin expression in SFEVs from the severe-OA group (Figure 1F,G, 2-way ANOVA *p* < 0.01), compared to EVs from the mild/moderate-OA patient group. OKS were significantly different between the two groups (*p* = 0.0443), supporting a distinction in patient-reported disease severity (Table 1).

### 3.3. Synovial Fluid and Synovial Fluid EVs Induce Distinct Transcriptomic Profiles in Human Articular Chondrocytes

To determine the functional effect of SFEVs, articular chondrocytes were cultured in the presence of either SFEVs (5 × 10^9^) or synovial fluid (4–8 μL/mL culture media) containing the equivalent amount of EVs from patients with mild/moderate OA or patients with severe OA, or left untreated for a period of 24 h, before being subjected to bulk RNA sequencing analysis (Figure 2A).

RNA sequencing analysis revealed that both synovial fluid and SFEVs profoundly affected the transcriptome of human OA articular chondrocytes but induced distinct profiles (Figure 2B–E; Appendix A). Comparing synovial fluid-treated and untreated chondrocytes, differential expression analysis identified 708 differentially expressed genes (DEGs) of log2FC > 0.58, of which 376 were upregulated, and 332 were downregulated (FDR < 0.05). The majority (91.6%) of these DEGs were protein-coding genes, with long non-coding RNAs representing 7.5%, and the remaining ~1% of genes representing microRNA, pseudogenes, and snoRNAs (Figure 2F). The DEGs included genes that regulate remodelling of cartilage extracellular matrix (ECM), e.g., *MMP13* (FC = +2.8), *MMP1* (FC = −2.5), *MMP3* (FC = −2.6), *ADAMTS1* (FC = −1.7), Collagen Triple Helix Repeat Containing 1 (*CTHRC1*; FC = +2.6), Periostin (*POSTN*; FC = +1.8), Tissue Inhibitor of Metalloproteinases 3 (*TIMP3*; FC = −1.7) and Tissue Factor Pathway Inhibitor 2 (*TFPI2*; FC = −3.0), and genes that mediate inflammation, including IL-1 receptor antagonist (*ILRN*; FC = −9.6), *TAC1* (Substance P; FC = −5.4) and CXCL8 (FC = 6.1) (Figure 2C,E).

Comparing SFEV-treated and untreated chondrocytes, differential expression analysis identified 328 DEGs with log2FC > 0.58 (FDR < 0.05), of which 121 were upregulated, and 207 were downregulated. The majority (95.4%) of these DEGs were protein-coding genes, with the remaining 4.6% of DEGs representing long non-coding RNAs (Figure 2F). Amongst the differentially expressed genes were several known mediators of cartilage matrix remodelling and inflammatory responses including the upregulation of *ANGPTL7* (FC = +3.0), *CTRHC1* (FC = + 1.7), *CXCL6* (FC = +3.9), *CXCL1* (FC = +1.6), *CXCL3* (FC = + 1.5), *MMP7* (FC = +2.6), *DUSP4* (FC = +2.5), *ITGA5* (FC = +2.0), and *OMD* (osteomodulin; FC = +2.7), and downregulation of *HAS3* (Hyaluronan Synthase 3; FC = −2.9), *ADAMTS12* (FC = −1.5), *ADAMTS14* (FC = −2.1), *FGF9* (FC = −2.4), which regulates chondrogenesis, and *HS3ST1* (FC = −2.7), a glycosaminoglycan which modulates matrix integrity (Figure 2E). Furthermore, several neuronal signalling mediators were differentially dysregulated, including *SEMA3A* (Semaphorin A, FC = −1.7), *PTGS2* (FC = −1.8) and *NGF* (FC = −1.5). Of note, comparing the DEGs in synovial fluid-treated chondrocytes and SFEV-treated chondrocytes, 48 upregulated DEGs and 59 downregulated DEGs were common to both (Figure 2E).

Next, we used Ingenuity Pathways Analysis (IPA) to understand the effects of both synovial fluid and SFEVs on chondrocyte cellular function. Firstly, to identify candidate upstream regulators (USRs) of the DEGs, we performed an upstream regulator analysis of the DEGs from synovial fluid-treated chondrocytes. In total, we identified >290 USRs that were predicted to be either activated (z-score ≥ 2) or inhibited (z-score ≤ −2) with significance of *p* < 0.01 (Figure 2G; Appendix A). Focusing on the most significant USRs, these could be classified into those that regulate (i) immune and inflammatory responses (e.g., *TNFA*), (ii) cartilage and ECM remodelling (e.g., *HOXA5*), or (iii) neuronal signalling and cellular communication (e.g., *NTRK1*). Notably, the majority of pro-inflammatory USRs, including *TNFA*, *IL-1A*, *IL-1B*, as well as multiple components of NFκB, were predicted to be inhibited, whilst the majority of anti-inflammatory USRs, including *IL-10* and corticosterone, were predicted to be activated (Figure 2G,H). In alignment with these USRs, pathway analysis identified significant dysregulation in canonical signalling pathways related to immune and inflammatory responses (e.g., “IL-10 signalling” and “Granulocyte adhesion and diapedesis”), cartilage and ECM remodelling (e.g., “Collagen degradation” and “Inhibition of MMPs”), and neuronal signalling and cellular communication (e.g., axonal guidance signalling) (Figure 2I; Appendix A).

Upstream regulator analysis of the DEGs from SVEV-treated chondrocytes identified only 18 USRs with z-scores ≥ 2 or ≤−2, with significance of *p* < 0.01, which could also be classified into those regulating (i) immune and inflammatory responses, (ii) cartilage and ECM remodelling, or (iii) neuronal signalling and cellular communication (Figure 2J; Appendix A). However, in comparison to the USRs of synovial fluid-treated chondrocytes, the SFEV USRs of SFEV-treated chondrocytes indicated a greater balance between pro-inflammatory/pro-catabolic mediators and anti-inflammatory/pro-anabolic mediators. For example, there was predicted activation of the pro-inflammatory *IL-6* (Figure 2K), as well as activation of the pro-catabolic mediators *OSCAR* (osteoclast-associated receptor) and *RARA* (retinoic acid receptor alpha), alongside the predicted activation of the anti-inflammatory *NR3C1* and the pro-anabolic Collagen Type II (complex). Pathway analysis of the DEGs from SFEV-treatment of chondrocytes identified significant dysregulation in pathways, which were similarly related to (i) immune and inflammatory responses (e.g., “IL-17 signalling” and “Eicosanoid signalling”), (ii) cartilage and ECM remodelling (e.g., “chondroitin sulphate biosynthesis” and “Glycosaminoglycan metabolism”), and (iii) neuronal signalling and cellular communication (e.g., “Axonal guidance signalling” and “Synaptogenesis signalling pathway”) (Figure 2L; Appendix A).

### 3.4. Synovial Fluid EVs from Patients with Severe OA Elicit Differential Catabolic and Inflammatory Transcriptomic Responses in Human Articular Chondrocytes

Next, we examined whether SFEVs derived from patients with severe OA induced distinct transcriptomic changes in chondrocytes compared to SFEVs from patients with mild/moderate OA (Figure 3A,B; Appendix A). Treatment with SFEVs from mild/moderate OA resulted in 273 DEGs (Log2FC > 0.58; FDR < 0.05), of which 172 were downregulated, and 101 were upregulated relative to untreated chondrocytes. In contrast, treatment with SFEVs from patients with severe OA yielded 297 DEGs, comprising 189 downregulated and 108 upregulated genes (Figure 3C). Among these, 202 DEGs were common to both treatment conditions, of which 128 were downregulated, and 74 were upregulated. As expected, these common genes included many of the genes previously identified as differentially expressed upon SFEV treatment of chondrocytes, e.g., the upregulation of *OMD*, *ANGPTL7*, *CTHRC1*, *CXCL1*, *CXCL3*, *DUSP4* and *ITGA5*, and the downregulation of *HAS3*, *ADAMTS12*, *ADAMTS14*, *FGF9*, *HS3ST1*, *SEMA3A*
*NGF*, and *PTGS2*.

With our defined cutoffs (Log2FC > 0.58, FDR < 0.05), a total of 71 DEGs (26%) were specific to treatment using SFEVs from mild/moderate OA, while 104 DEGs (35%) were specific to treatment with SFEVs from severe OA (Figure 3C). Focusing on the 104 genes (70 downregulated, 34 upregulated) that were preferentially dysregulated by severe OA SFEVs, several were key regulators of cartilage and ECM remodelling, including *CRTAC1* (Cartilage Acidic Protein 1, FC = +1.7), *RUNX2* (FC = +1.7), *COL6A3* (FC = +1.5), *TNC* (Tenascin-C, FC = +1.6), a matricellular protein increased in cartilage injury and in OA, *SOX9* (FC = −1.5), *GDF5* (FC = −1.5), and *MATN2* (Matrilin2, FC = −1.5). In addition, genes specifically altered by severe SFEVs also included several central regulators of immune and inflammatory responses, e.g., *CXCL5* (FC = +1.6), *NFKBIZ* (FC = +1.7), and the neuronal, signalling inflammatory mediators *NRG1* (Neuregulin 1, FC = −1.5), *CHN1* (Chimerin 1, FC = +1.6), and *SYT7* (Synaptotagmin 7, FC = −1.6) (Figure 3D).

Assigning the DEGs to canonical signalling pathways identified multiple significant dysregulated pathways with both mild/moderate-SFEVs and severe-SFEV treatments, which regulate (i) immune and Inflammatory responses, (ii) cartilage and ECM remodelling, and (iii) neuronal signalling and cellular communication (Figure 3E; Appendix A). Notably, chondrocytes treated with SFEVs from severe OA displayed more significant dysregulation, with greater numbers of DEGs aligned to pathways and lower LogP values. For example, the pro-inflammatory pathways “Eicosanoid signalling” “Agranulocyte adhesion and diapedesis”, and “osteoarthritis pathway”, and in neuroinflammatory pathways “neuroinflammation signalling pathway”, “RAR activation”, and “DHA signalling” (Figure 3E).

In line with the canonical pathway analysis, performing upstream regulator analysis revealed distinct differences in the predicted activation status of USRs between the mild/moderate OA SFEV-induced DEGs and the severe OA SFEV-induced DEGs. The predicted USRs and their assigned activation status of the mild/moderate OA SFEVs were strongly anti-inflammatory, with inhibition of the pro-inflammatory cytokines *IL-2* and *IL-13*, and activation of *NR3C1* and corticosterone (Figure 3F; Appendix A). In contrast, the predicted USRs and their activation status of the severe SFEVs was more strongly pro-inflammatory and pro-catabolic, with activation of *IL-6*, *OSCAR* and Col Type II (complex) (Figure 3G; Appendix A).

### 3.5. Synovial Fluid EVs from Patients with Severe OA Induce the Differential Release of Catabolic and Inflammatory Mediators in Human Articular Chondrocytes

Having observed that SFEVs elicited differential effects on the chondrocyte transcriptome relating to pathways that mediate inflammation and cartilage catabolism, we next examined if these effects extended to the protein release of known catabolic and inflammatory mediators. To this end, we used Luminex and ELISA assays to profile the protein secretion of a panel of mediators (including MMPs, pro-inflammatory cytokines, and neurotrophins) in cells exposed to either SFEVs from severe OA or from mild/moderate OA, compared to untreated chondrocytes (Figure 4A).

Analysis of protein release of matrix metalloproteases (MMPs) revealed that chondrocytes treated with SFEVs from patients with severe OA exhibited a significant increase in the secretion of MMP3 (*p* < 0.001), whereas no such increase was observed in cells treated with SFEVs from mild/moderate OA. Both SFEVs from severe and mild/moderate OA induced a significant increase in MMP13 secretion; however, SFEVs from severe OA induced a significantly greater effect (*p* < 0.05) compared to those from mild/moderate OA. Finally, compared to untreated cells, neither SFEVs from mild/moderate OA or severe OA induced any significant change in the secretion of MMP1, albeit the levels of MMP1 were significantly greater (*p* < 0.05) in chondrocytes subjected to severe SFEVs compared to mild/moderate SFEVs (Figure 4B). Furthermore, although not reaching statistical significance, the secretion of the endogenous inhibitor of MMPs, TIMP3, was on average reduced in chondrocytes exposed to SFEVs, particularly those isolated from the synovial fluid of patients with severe OA.

Investigation of the secretion of pro-inflammatory mediators revealed significantly increased secretion of the pro-angiogenic ICAM1 (*p* < 0.001) in chondrocytes exposed to SFEVs from severe OA, but no effect of SFEVs from mild/moderate OA, compared to untreated control. Secretion of IL-6 was significantly increased in chondrocytes treated with mild/moderate SFEVs (*p* < 0.001) and further significantly increased (*p* < 0001) in cells exposed to severe SFEVs (Figure 4B).

Finally, we observed a significant increase in the release of neuronal signalling mediators from chondrocytes subjected to SFEVs. Firstly, the secretion of TAC1 (substance P), was markedly increased (~18-fold) in chondrocytes treated with SFEVs from mild/moderate OA (*p* < 0.05). Notably, we observed a comparable (~18-fold) increase in TAC1 in cells exposed to severe OA SFEVs, compared untreated control, although this did not reach significance. Secondly, NGF secretion was increased in cells treated with severe OA SFEVs, but not from patients with mild/moderate OA, compared to untreated control cells (Figure 4B).

Retrospective analysis of the RNAseq data of these specific mediators revealed mRNA expression data that was broadly in agreement with protein secretion levels observed for MMP1, MMP3, MMP13, TIMP3, and ICAM1. However, there was no change in *IL-6* or *TAC1* transcript levels in chondrocytes subjected to SFEVs, and a reduction in *NGF* mRNA in chondrocytes treated with SFEVs from either mild/moderate or severe OA (Figure 4C), suggesting that SFEVs mediate these particular pro-inflammatory neuronal mediators post-transcriptionally.

Finally, we utilised a customised Luminex panel to assess whether specific protein cargo within SFEVs (MMP1, MMP3, MMP13, NGF, BDNF, IL-6, and IL-1β) could be detected and, furthermore, determine the relationship between analyte concentration/EV particle and patient-reported outcome measures (VAS, EQ5D, and OKS). Lysates were prepared from EVs isolated from 13 OA patients (7 mild/moderate, 4 severe, and 2 unclassified). Group-based comparisons were conducted using the 11 patients with complete classification data. Importantly, all targeted analytes were detectable within the SFEV lysates (Figure 4D), although absolute concentrations detected were between 4-fold and 35-fold lower than that found in the synovial fluids (Appendix A). Notably, MMP3 was only qualitatively detected, as its concentration exceeded the upper limit of the standard curve. This finding suggests that SFEVs transport key pro-catabolic and neuro-inflammatory factors within the OA synovial joint. Among the detected analytes, NGF and MMP1 were the most highly enriched in SFEVs compared to equivalent synovial fluid concentration, suggesting these proteins may be preferentially packaged into SFEVs (Appendix A). Moreover, we observed a significant positive correlation between SFEV NGF concentration/EV particles and VAS pain severity scores (*p* < 0.05, Figure 4E), suggesting a potential mechanistic link between SFEVs and patient-reported pain perception in OA.

## 4. Discussion

This study provides evidence for a functional relationship between the characteristics of synovial fluid EVs (SFEVs), chondrocyte phenotype, and OA severity. For the first time, we demonstrate that size, concentration, and tetraspanin marker expression of SFEVs are related to OA severity. Furthermore, we demonstrate that SFEVs from patients with severe OA drive a distinct catabolic inflammatory articular chondrocyte phenotype.

We have been able to categorise patients into mild/moderate- and severe-OA groups using patient-reported quality of life, pain, and joint function combined data. Synovial fluid EV were isolated and characterised. Interestingly, an inverse relationship was found between EV concentration and EQ5D, suggesting that lower EV abundance may be associated with more advanced disease or reduced joint function. One possible explanation is that synovial tissue in severe OA becomes fibrotic, inflamed, or otherwise dysfunctional, leading to altered EV production or release. However, this remains speculative, as our study did not directly assess tissue pathology or EV biogenesis mechanisms.

Although patients were not selected based on inflammatory biomarkers, the cohort studied likely represents an inflammatory OA endotype—all patients exhibited synovitis by MRI, assessed across 11 compartments of the knee, and all patients were classified as having moderate or severe synovitis. Importantly, synovitis scores were not significantly different between the mild/moderate- and severe-OA groups, indicating that inflammation was not a driving factor in the stratification but was a common feature across the cohort. This inflammatory background may help explain the strong pro-inflammatory responses observed in chondrocytes treated with SFEVs.

Synovial inflammation in OA is considered a driver of both cartilage degeneration and pain sensitisation. Therefore, our finding that SFEVs can promote the expression and release of not only cartilage catabolic mediators including *CRTAC1*, *TNC*, *COL6A3*, MMP3, and MMP13, but also pro-inflammatory mediators such as TAC1 (Substance P), CXCL5, and IL-6 in articular chondrocytes suggests that SFEVs may contribute to both cartilage degeneration and pain sensitisation in the joint. Notably, the induction of many of these pro-catabolic and pro-inflammatory effects was significantly greater in chondrocytes subjected to SFEVs from more severe OA, compared to SFEVs from mild/moderate OA, aligning with the previous study by Zhang et al. (2023) [18], who reported that EVs in severe OA contain pro-inflammatory cytokines that exacerbate joint damage. Furthermore, potentially compounding the pro-catabolic effect of SFEV-induced MMPs on cartilage, we also found that SFEVs reduced the expression of the tissue inhibitor of MMPs, *TIMP3*. Reduced *TIMP3* expression may lead to unchecked MMP activity, tipping the balance towards cartilage matrix degradation. These findings suggest that the severe-OA group in our study likely reflects an inflammatory endotype, as evidenced by the SFEV-induced upregulation of pro-inflammatory (e.g., IL-6, CXCL5, ICAM1) and nociceptive (e.g., NGF, TAC1) mediators in chondrocytes.

In addition to the induction of these well-established pro-inflammatory catabolic mediators, pathway analysis of the RNA sequencing transcriptomic data revealed that SFEVs modulated a number of canonical inflammatory signalling pathways that regulate immune and inflammatory responses, cartilage and ECM remodelling, and neuronal signalling. For example, chondrocytes subjected to SFEVs exhibited increased expression of *DUSP4*, which is a known regulator of MAPK inflammatory signalling in chondrocytes [19], and *ITGA5*, a receptor for fibronectin, supporting the potential for SFEVs to shape the inflammatory joint microenvironment. Recently, ITGA5+ synovial fibroblasts were shown to exacerbate inflammatory joint pathology in a collagen-induced arthritis model [20].

We also found that SFEVs mediated transcriptional changes in several genes that mediate vascular biology. Although the role of angiogenesis and neovascularisation in OA is not fully understood [21,22], angiogenesis has been reported as a feature of OA progression, driven by chronic inflammation and oxidative stress [22,23]. Unlike healthy cartilage, which is avascular, OA cartilage undergoes aberrant neovascularization [23], a process facilitated by the induction of pro-angiogenic factors such as VEGF [24,25] and ICAM1, which is induced in chondrocytes by IL-1β and found in areas of damaged cartilage [26]. The presence of new vasculature is thought to exacerbate inflammation and pain by supplying nutrients and inflammatory mediators to the joint. Here, we observed no change in expression of *VEGF* with SFEVs from either mild/moderate OA or severe OA. However, SFEVs from patients with severe OA induced increased expression of *ANGPTL7* and *SOD2*, and increased expression and secretion of *ICAM1* in articular chondrocytes. *ANGPTL7*, a member of the angiopoietin-like family, promotes angiogenesis. Previously, it was reported that the expression of *ANGPTL7* in the joint is induced by mechanical stimuli, whilst functionally its over-expression promotes chondrocytes proliferation and calcification [27]. SOD2 is an antioxidant enzyme that regulates reactive oxygen species (ROS) production [28], which is known to promote angiogenic and inflammatory signalling [29,30]. Interestingly, the ablation of *SOD2* in models of mechanical joint loading promotes cartilage degeneration, and its expression, similarly to other superoxide dismutases (e.g., *SOD1* and *SOD3*), is downregulated in human OA cartilage [31,32] and decreases during disease progression in a spontaneous OA animal model [31]. Notably, previous studies have reported that EVs can transport pro-angiogenic factors [33], supporting the potential role of SFEV in actively contributing to neovascularization, and our findings here would support that concept.

Increasing evidence suggests that peripheral pain sensitisation in OA is associated with synovitis [7,34,35] and, in part, mediated by the interactions between chondrocytes, immune cells, and synovial fibroblasts within the synovial tissue [7], leading to the production of cytokines and growth factors that are capable of sensitising nociceptors directly or activating neuronal signalling pathways [36,37,38].

Notably, we found that the secretion of NGF protein was significantly increased in chondrocytes stimulated with SFEVs from patients with severe OA. Furthermore, TAC1 (Substance P) secretion was increased approximately 18-fold in chondrocytes exposed to SFEVs from either mild/moderate or severe OA. NGF is a well-established mediator of OA pain which, via binding to TrkA, sensitises peripheral nociceptors, promotes sprouting, and drives chronic pain states within the joint [39]. Indeed, NGF inhibition has been shown to significantly reduce pain in OA models and in clinical trials [40,41]. Similarly, Substance P, a neuropeptide encoded by *TAC1*, plays a pivotal role in neurogenic inflammation and pain signalling by activating neurokinin-1 (NK1) receptors on sensory afferent neurons [42]. At elevated levels in OA synovial fluid [43], NK1-receptor antagonists show analgesic efficacy in rodent models of arthritis [44], and *TAC1* SNPs are associated with symptomatic knee OA pain [45]. Thus, taken together, these findings suggest that SFEVs within the OA joint may contribute to the establishment and amplification of pain signalling networks in cartilage through the induction of NGF and Substance P.

Collectively, SFEVs from patients with severe OA elicited more pronounced effects on chondrocytes than SFEVs from patients with mild/moderate OA. Indeed, upstream regulator analysis revealed a striking divergence in the predicted upstream mediators of SFEV action depending on OA severity. The chondrocyte transcriptome induced by SFEVs from patients with mild/moderate OA was associated with the upstream activation of corticosterone, a potent anti-inflammatory mediator, suggesting that in less severe disease, SFEVs may help maintain a homeostatic balance, supporting both anabolic and anti-inflammatory responses in chondrocytes. This contrasts with SFEVs from patients with severe OA, where IL-6, a well-known pro-inflammatory cytokine, emerged as a key activated upstream regulator of the chondrocyte transcriptome, aligning with the broader pro-inflammatory and catabolic gene and protein expression profiles we observed.

Together, these data suggest that in healthy or mildly affected joints, SFEVs might play a regulatory role, modulating chondrocyte responses in a way that resembles the homeostatic functions of healthy synovial fluid. However, as OA progresses to a severe stage, this balance appears to be shifted towards promoting inflammation and matrix degradation, potentially exacerbating cartilage degeneration and pain. However, it should be noted that these effects were markedly different to the effect of synovial fluid on chondrocytes. This suggests that although SFEVs likely contribute to driving inflammation and cartilage degeneration, additional non-EV encapsulated synovial fluid factors have a predominant effect on chondrocytes. Although the inclusion of an EV-depleted synovial fluid control would have strengthened this comparison, reliable depletion was not feasible due to the high viscosity of synovial fluid. Methods that could remove EVs (e.g., repeated ultracentrifugation or size exclusion) would have altered the composition of the fluid, necessitating additional control groups and complicating interpretation. Therefore, full synovial fluid was used to maintain comparability and physiological relevance.

Further supporting the role of SFEVs as active mediators of OA pathology, targeted protein analysis revealed that key catabolic (MMP1, MMP3, MMP13) and neurotrophic (NGF, BDNF) factors were detectable within SFEV lysates following detergent lysis. Notably, NGF and MMP1 were significantly enriched in SFEVs compared to whole synovial fluid, suggesting that these factors may be selectively packaged into EVs rather than freely circulating in the synovial fluid joint environment. This targeted enrichment highlights a potential role for SFEVs in the direct delivery of pro-catabolic and neuro-inflammatory signals to chondrocytes and other joint-resident cells, which may help explain their strong functional effects. Moreover, the observed positive correlation between SFEV NGF concentration and patient-reported pain severity (VAS scores) provides further evidence that SFEVs may contribute to peripheral pain sensitisation in OA. Given that NGF plays a key role in OA pain via sensitization of nociceptive neurons, these findings raise the intriguing possibility that SFEVs serve as vehicles for nociceptive signalling, actively influencing pain perception in severe OA.

Several limitations should be considered in this study. First, the isolation method used for EVs from synovial fluid was ultracentrifugation, which, although widely used, may not be ideal as it can co-precipitate other molecules, such as lipids. This is a common issue with many EV isolation techniques, and while it could affect the purity of the EV fraction, ultracentrifugation remains one of the most effective methods currently available. Due to the small sample volumes available, the contamination of non-EV proteins and lipids was not assessed in this study. Another limitation is that our study did not fully profile EV cargo, as the volume of synovial fluid limited this possibility. Future work should include proteomics and small RNA sequencing. Previous studies have reported that EVs from both the infrapatellar fat pad [11] and synovial fluid [18] in OA patients, particularly those with advanced disease, contain a higher concentration of pro-inflammatory cargo. Therefore, such analyses, together with EV surface marker expression, could lead to the identification of candidate signatures as biomarkers of OA severity or help to stratify OA patient molecular endotype, as well as provide more mechanistic understanding of the EV mediated functional effects on articular chondrocytes reported here. It should also be noted that we categorised OA severity based on patient-reported outcomes, including the well-established Oxford Knee Score [46,47] and EQ5D [48]. These instruments collectively assess joint mobility, function, pain, and quality of life—aspects that are clinically meaningful and directly relevant to patients. However, given the partial disconnect between patient-reported outcome measures and structural pathology in OA [49], future studies applying complementary stratification methods, such as MRI-based assessments or molecular biomarker profiling, would help to further validate EV-associated phenotypes and disease severity classifications. Additionally, although patients were stratified into discrete groups for clarity, OA exists along a continuum. Future analyses could explore whether patients who did not meet our severity thresholds exhibit intermediate or distinct SFEV characteristics. The relatively small sample size in this study, a consequence of using human samples, is an inherent limitation. Future studies with larger sample sizes would be beneficial to confirm and extend these findings. Finally, future studies incorporating histological analysis of cartilage and synovium could further validate EV-associated inflammatory or vascular signatures and help refine OA endotype classification.

## 5. Conclusions

In summary, SFEVs, particularly those from patients with severe OA, drive a pro-inflammatory and pro-catabolic phenotype in OA articular chondrocytes. These findings provide further evidence for the role of synovial inflammation in exacerbating cartilage degeneration and for the rationale of therapeutically targeting SFEVs to disrupt the intracellular inflammatory and degenerative signalling cascades within the OA joint.

## Figures and Tables

**Figure 1 biomolecules-15-00829-f001:**
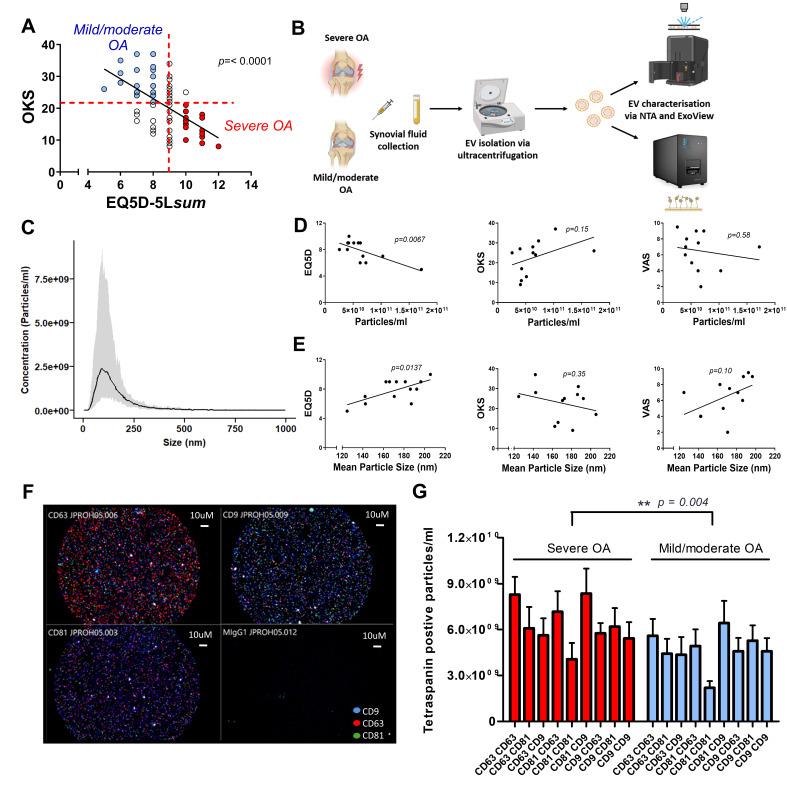
(**A**) Stratification of patients into “mild/moderate OA” and “severe OA” groups based on linear regression analysis between EQ5D and Oxford Knee Score (OKS) (*p* < 0.0001, r^2^ = 0.3287). (**B**) Schematic of the experimental workflow. Synovial fluid was collected from patients with severe OA or mild/moderate OA, and extracellular vesicles (EVs) were isolated by ultracentrifugation. EVs were characterised using nanoparticle tracking analysis (NTA) and ExoView. (**C**) Histogram showing the average and size distribution of synovial fluid extracellular vesicles (SFEVs) isolated by centrifugation, as measured by NTA (*n* = 13 patients, *n* = 7 mild/moderate OA, *n* = 4 severe OA, *n* = 2 excluded from either group category). (**D**) Linear regression analysis of EV concentration (particles/mL) measured by NTA with EQ5D, OKS, and VAS patient-reported scores (*n* = 13 patients). (**E**) Linear regression of SFEV mean particle size measured by NTA with EQ5D, OKS, and VAS scores (*n* = 13 patients). (**F**) Representative ExoView fluorescence image of SFEVs from a mild/moderate OA patient (dilution 1:500). Tetraspanins shown: CD9 (blue), CD63 (red), CD81 (green). (**G**) Two-way ANOVA analysis of tetraspanin expression on SFEVs from patients with severe OA and mild/moderate OA, as measured by ExoView (*p* = 0.004, *n* = 11). Tetraspanin-positive EV counts detected on ExoView chips functionalised with CD9, CD63, or CD81 capture antibodies and stained with corresponding fluorescent detection antibodies. EVs from patients with mild/moderate and severe OA were analysed at 1:500 dilution. Despite lower EV concentrations in the severe-OA group (as shown in (**D**)), ExoView revealed a higher number of tetraspanin-positive EVs in this group, suggesting enrichment of tetraspanin expression.

**Figure 2 biomolecules-15-00829-f002:**
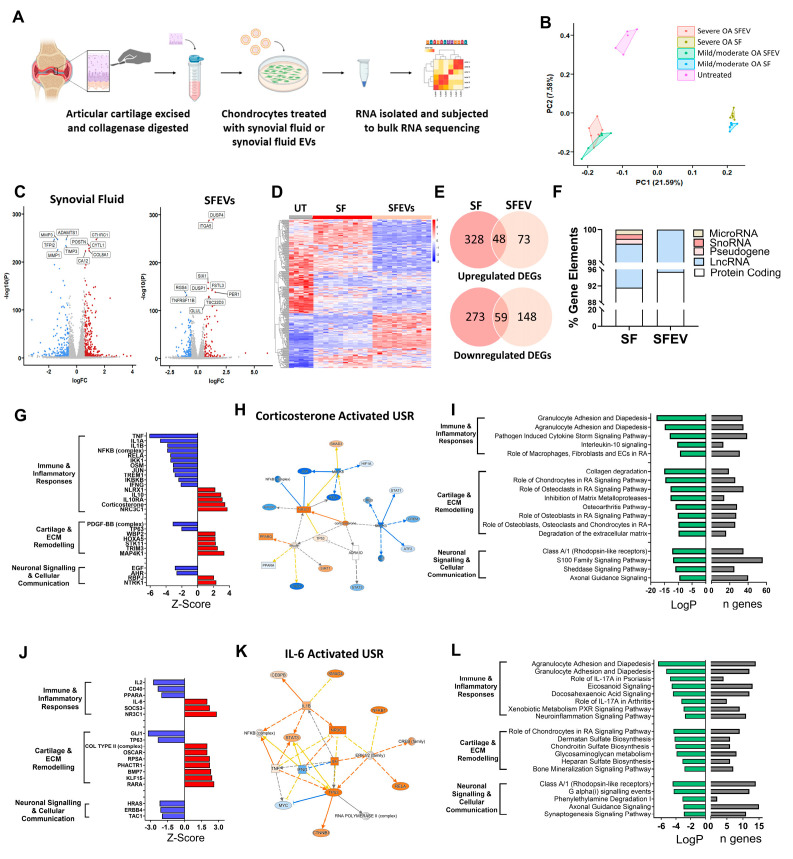
(**A**) Schematic of experimental workflow. OA cartilage was collagenase digested. Isolated chondrocytes were cultured for 24 h with either synovial fluid (SF) or isolated SFEVs (*n* = 6) from the same patients or left untreated. Chondrocytes were treated with either 5 × 10^9^ SFEV particles per well (48 well plate) or with synovial fluid volumes containing an equivalent number of EVs: 4.5 μL per well for severe OA and 8 μL per well for mild/moderate OA, based on measured EV concentration and isolation efficiency. Total RNA was extracted and subjected to bulk RNA sequencing. (**B**) Principal Component Analysis (PCA) plot demonstrating distinct clustering of samples following RNA sequencing. (**C**) Volcano plots highlighting DEGs (FDR < 0.05, FC > 1.5) in SF-treated vs. untreated and SFEV-treated vs. untreated chondrocytes. (**D**) Heatmap displaying the DEGs between SF-treated and SFEV-treated and untreated chondrocytes. (**E**) Venn diagrams illustrating the common and unique DEGs of SF-treated and SFEV-treated vs. untreated chondrocytes. (**F**) Percentage distribution of gene elements of DEGs identified in SF-treated and SFEV-treated chondrocytes. (**G**) Z-scores of identified upstream regulators of the transcriptome induced by SF-treated chondrocytes, as identified by IPA. Positive z-scores ≥ 2 represent “activated regulators”; negative z-scores ≤ −2 represent “inhibited regulators”. (**H**) Molecular network map of the activated upstream regulator corticosterone with connections to DEGs in SF-treated chondrocytes. (**I**) Top significantly dysregulated canonical signalling pathways in SF-treated chondrocytes as identified by IPA, showing LogP significance and number of DEGs (n) in the dataset aligned to each pathway. (**J**) Z-scores of identified upstream regulators of the transcriptome induced by SFEV-treated chondrocytes, as identified by IPA. Positive z-scores ≥ 2 represent “activated regulators”; negative z-scores ≤ −2 represent “inhibited regulators”. (**K**) Molecular network map of the activated upstream regulator IL6 with connections to DEGs in SFEV-treated chondrocytes. (**L**) Top significantly dysregulated canonical signalling pathways in SFEV-treated chondrocytes as identified by IPA, showing LogP significance and number of DEGs.

**Figure 3 biomolecules-15-00829-f003:**
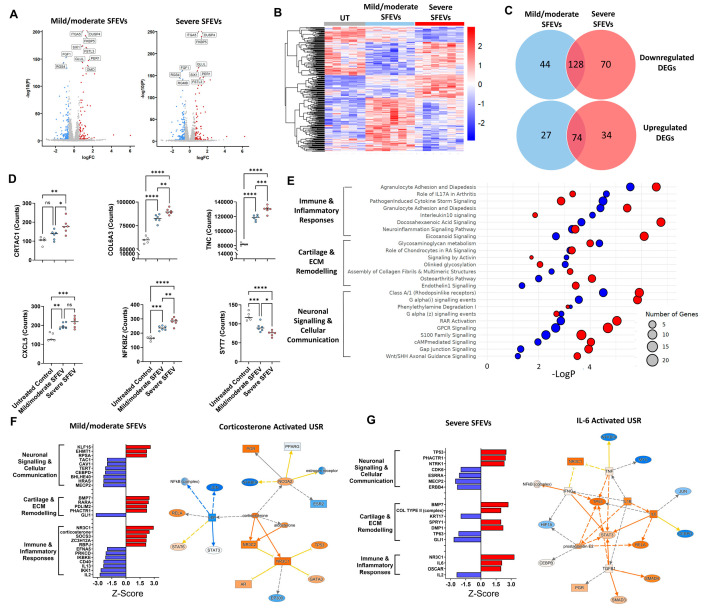
(**A**) Volcano plots highlighting DEGs (FDR < 0.05, FC > 1.5) in chondrocytes treated with mild/moderate SFEV vs. untreated and treated with severe SFEV vs. untreated. (**B**) Heatmap displaying differential transcriptome of untreated, mild/moderate-SFEV-treated, and severe-SFEV-treated chondrocytes. (**C**) Venn diagrams illustrating the common and unique DEGs of mild/moderate-SFEV-treated and severe-SFEV-treated chondrocytes. (**D**) Normalised counts from bulk RNA sequencing for *CRTAC1*, *COL6A3*, *TNC*, *CXCL5*, *NFKBIZ*, and *SYT7*. Data points represent individual sample values for untreated (*n* = 5), mild/moderate-SFEV-treated (*n* = 6), and severe-SFEV-treated (*n* = 6) conditions, with bar showing mean value. Statistical analysis was performed using one-way ANOVA with Tukey’s multiple comparisons post hoc tests (* *p* < 0.05, ** *p* < 0.01, *** *p* < 0.001, **** *p* < 0.0001). (**E**) Bubble plot of canonical signalling pathway enrichment for DEGs from mild/moderate-SFEV-treated (blue bubbles) and severe-SFEV-treated (red bubbles) chondrocytes, with −LogP significance represented on the x-axis and bubble size reflecting number of DEGs aligned to each pathway. (**F**) Z-scores of identified upstream regulators of the transcriptome induced by mild/moderate-SFEV-treated chondrocytes, as identified by IPA. Positive z-scores ≥ 2 represent “activated regulators”; negative z-scores ≤ −2 represent “inhibited regulators”. Molecular network map shows the activated upstream regulator corticosterone with connections to DEGs in mild/moderate-SFEV-treated chondrocytes. (**G**) Z-scores of identified upstream regulators of the transcriptome induced by severe-SFEV-treated chondrocytes, as identified by IPA. Positive z-scores ≥ 2 represent “activated regulators”; negative z-scores ≤ −2 represent “inhibited regulators”. Molecular network map shows the activated upstream regulator IL-6 with connections to DEGs in severe-SFEV-treated chondrocytes.

**Figure 4 biomolecules-15-00829-f004:**
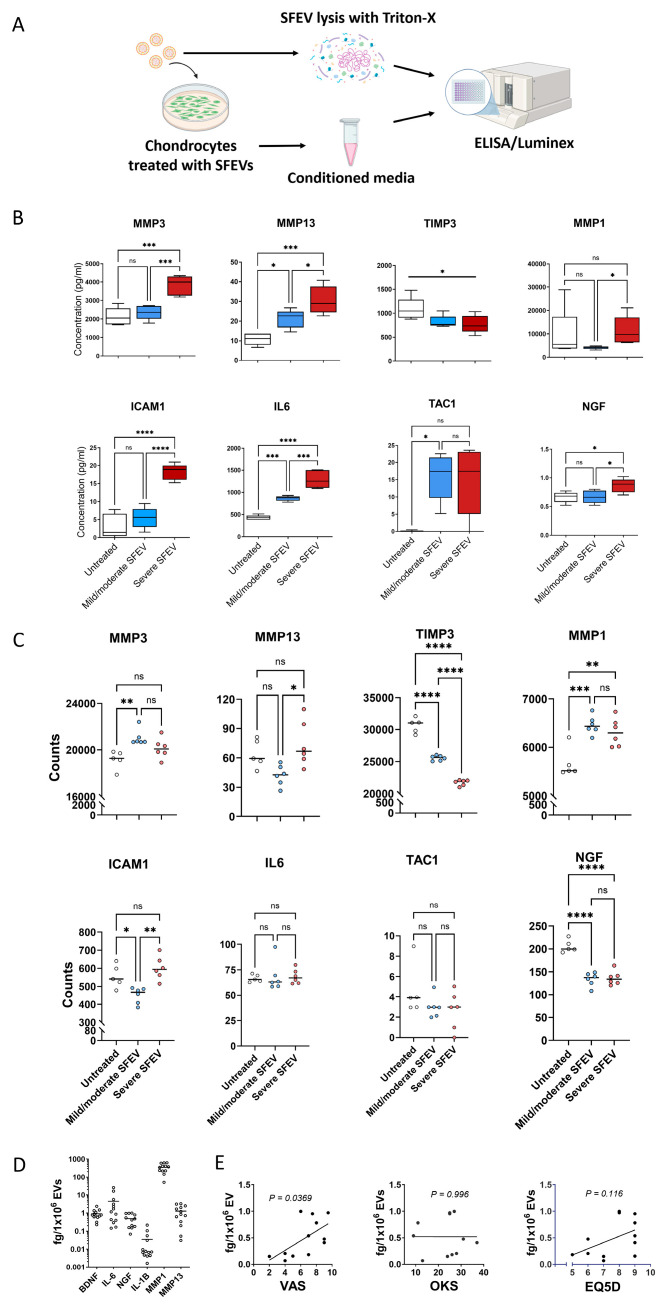
(**A**) Schematic of experimental design showing chondrocyte stimulation with SFEVs, collection of conditioned media for secreted protein analysis (ELISA and Luminex), and direct SFEV lysis for EV cargo profiling. EVs were isolated from *n* = 13 OA patients (7 mild/moderate, 4 severe, and 2 unclassified); group comparisons were performed using *n* = 11 classified patients. Primary human OA chondrocytes, isolated by collagenase digestion of articular cartilage were cultured for 24 h with SFEVs from patients with either mild/moderate or severe OA, or left untreated. Conditioned media was collected, and protein analyte concentrations quantified using Luminex (MMP1, MMP3, MMP13, IL-6, and NGF) or ELISA (TAC1, TIMP3, and ICAM1). Synovial fluid and SFEVs were lysed with Triton-X and protein analytes quantified using a customised Luminex (MMP1, MMP3, MMP13, BDNF, NGF, IL-6, and IL-1β). (**B**) Box and whisker plots showing protein concentrations (pg/mL) of analytes in untreated (*n* = 5), mild/moderate-SFEV-treated (*n* = 6), and severe-SFEV-treated (*n* = 6) chondrocytes, as measured by Luminex and ELISA. Boxes represent the interquartile range (IQR; 25th to 75th percentiles), with the line indicating mean and whiskers representing minimum–maximum values. (**C**) Normalised counts from bulk RNA sequencing for MMP1, MMP3, MMP13, TIMP3, ICAM1, IL6, TAC1, and NGF in untreated (*n* = 5), mild/moderate-SFEV-treated (*n* = 6), and severe-SFEV-treated (*n* = 6) chondrocytes. Statistical analysis was performed using one-way ANOVA with Tukey’s post hoc multiple comparison tests, * *p* < 0.05, ** *p* < 0.01, *** *p* < 0.001, **** *p* < 0.0001. (**D**) Scatter plots of the protein concentration (pg/mL) of the analytes BDNF, IL-6, NGF, IL1β, MMP1, and MMP13 in SFEV lysates. (**E**) Linear regression analysis of SFEV NGF protein concentration with OKS, VAS, and EQ5D patient-reported scores (*n* = 12 patients). NGF protein was quantified by ELISA of SFEVs lysed with 0.5% Triton-X. Data points represent individual sample values, and bars show the mean.

**Table 1 biomolecules-15-00829-t001:** Characteristics of OA patient cohort for SFEV analyses.

	Mild/Moderate OA	Severe OA	*p* Value
	*n* = 7	*n* = 4	
OKS ^1^	26.8 ± 3.3	15.25 ± 3.38	* *p* = 0.044
EQ5D(sum) ^2^	7.6 ± 0.48	8.750 ± 0.63	*p* = 0.172
VAS ^3^	6.6 ± 0.93	5.667 ± 1.86	*p* = 0.610
Synovitis ^4^	16.0 ± 1.07	16.5 ± 0.65	*p* = 0.698
Age	64.0 ± 6	57.5 ± 6	*p* = 0.503
Sex (male:female)	5:2	1:3	
BMI ^5^	32.2 ± 2.3	35.2 ± 2.5	*p* = 0.421

^1^ OKS—Oxford Knee Score. ^2^ EQ5Dsum—summation of scores across all 5 dimensions: mobility, self-care, usual activities, pain/discomfort and anxiety/depression. ^3^ VAS = visual analogue scale of pain severity. ^4^ Synovitis—assessed using a semiquantitative scoring system at 11 anatomical sites within the knee joint, following the method proposed by Guermazi et al. (2011) [15]. Each site was graded on a scale from 0 to 2 based on synovial thickness measured via contrast-enhanced MRI: grade 0 (<2 mm), grade 1 (2–4 mm), and grade 2 (>4 mm). The total synovitis score was calculated by summing the grades across all sites, yielding a maximum possible score of 22. ^5^ BMI = body mass index, weight (kg)/height (m)^2^. * *p* < 0.05, significant difference between groups. ± = standard error of the mean.

## Data Availability

Data is contained within the article or Appendix A.

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
