# Peer review of "Synovial Fluid Extracellular Vesicles from Patients with Severe Osteoarthritis Differentially Promote a Pro-Catabolic, Inflammatory Chondrocyte Phenotype"

_biomolecules, 2025, doi:10.3390/biom15060829_

Round 1

Reviewer 1 Report

Comments and Suggestions for Authors

The authors  aim  to investigate the role of synovial fluid extracellular vesicles (SFEVs) in osteoarthritis (OA) by:

Characterizing SFEVs from patients with different OA severities (mild/moderate vs. severe), in terms of size, concentration, and surface markers.

Evaluating how these SFEVs functionally affect human articular chondrocytes, particularly in driving pro-inflammatory and catabolic responses that contribute to cartilage degradation and joint pain.

The study seeks to determine whether SFEVs contribute to OA pathogenesis and pain by acting as mediators of cellular cross-talk within the inflamed joint environment.

Overall this is a this is a solid and insightful contribution to OA research with novel findings on the inflammatory role of SFEVs in cartilage degeneration and pain.

Some aspects needs to be addressed: 

No remarks regarding title, abstract or introduction.

Methods:

line 80/81 it is mentioned that the tissue and SF was collected from OA patients undergoing joint surgery. I assume these were al joint replacement surgeries, please clarify.

How was the SF collected, by punction or when the joint was opened and how much was collected. 

line 87: From which compartment was the cartilage harvested, femur, tibia, or was a mix used. Was one donor used for OA chondrocyte culture or different donors?

line 145-150: Why was the cut-off of 21 used while it is often seen that the cut-of is 20? Similar why were these specific cut-offs chosen for the EQ5D. What was the amount of patients not fitting one of these groups.

As there is no control how specific these findings are (yes there is difference between mild vs severe based on these criteria) it would be helpful to see if the patients not fulfilling these criteria have also a different SFEVprofile, please discuss. 

Results:

I don't understand the n-values presented in table 1 or specific experimentsts. Why are the data of only 7 mild patients and 4 severe patients presented while in line 172 13 patients (mild) are mentioned.  It is remarkable that the age of the severe patients is relatively young. This suggests that this is a fast progressive group of patients which may interfere with the profile you suggest. Why is an asterisk presented in the table? To highlight  the significance of OKS difference?  Please clarify this in the text 

Line 196: does it matter which amount of EVs or synovial fluid is added (is there data available) and why is there a range used for synovial fluid. On what data is this based?

discussion: 

line 479-480 although indeed neovascularization is demonstrated in more severe patients this is not as black and white as stated here. Especiallly as this is not assessed in this study, this section feels a bit speculative.

General remarks: In the introduction and discussion the emphasis is on the inflammatory OA patient; what makes the donors used for EV isolation be these type of patients?

Why was histology not considered as assessment of cartilage severity and/or neovascularization? Similar for synovial tissue histology which might provide data on inflammatory status.

Author Response

Comments 1: line 80/81 it is mentioned that the tissue and SF was collected from OA patients undergoing joint surgery. I assume these were al joint replacement surgeries, please clarify.

Response 1: Thank you for your observation. Yes, all synovial fluid and cartilage samples were obtained from patients undergoing total knee replacement surgery due to osteoarthritis. We have clarified this in the Methods section.
“Articular cartilage and synovial fluid were collected from OA patients undergoing total knee replacement surgery...”

Comments 2: How was the SF collected, by punction or when the joint was opened and how much was collected. 

Response 2: Thank you for your question. Synovial fluid was collected intra-operatively during joint surgery, immediately after the joint capsule was opened. Collection volumes varied between patients, ranging from approximately 100 µL to 8 mL, depending on availability and the amount the surgeon was able to aspirate. This has now been clarified in the Methods section.
“Synovial fluid was collected intra-operatively upon joint capsule opening, with volumes ranging from ~100 µL to 8 mL depending on fluid availability.”

Comments 3: line 87: From which compartment was the cartilage harvested, femur, tibia, or was a mix used. Was one donor used for OA chondrocyte culture or different donors?

Resposne 3: Thank you for your comment. In order to harvest sufficient intact articular cartilage for the generation of OA chondrocytes, a mixture of tibilal plateau and femoral condyle compartments were used.  For all experiments involving OA chondrocyte cultures, cartilage from a single donor was used to avoid inter-donor variability and preserve individual phenotypic responses. The Methods section has been updated to include this information.
Cartilage was obtained from macroscopically intact regions of the joint, including from the femoral condyles and tibial plateau. Fresh cartilage was dissected into 1-3mm3 pieces, and digested using 2mg/ml collagenase (Merck, C9891) in DMEM (Merck, D6429) for 7 hours at 37˚C. Digested cartilage was filtered through a 70µm cell strainer, the resultant filtrate centrifuged at 400 g for 5 min, and then resuspended and cultured in chondrocyte growth media (DMEM, 10% FCS, 1% NEAA, 2 mM L-glutamine (ThermoFisher Scientific, Gloucester, UK, 25030024), 1% penicillin and streptomycin (100 U/mL penicillin and 100µg/mL streptomycin) (Merck, P4333), 2.5ug/ml amphotericin B (Merck, A2942)). Growth media was replaced every 3–4 days and cells were passaged upon reaching 70% confluency. Chondrocytes were cultured in standard monolayer and used at early passage (P2) to minimise dedifferentiation. The same donor-derived chondrocyte population (BMI: 28.36) was used across all treatment conditions to minimise inter-donor variability. No data were collected on OKS, EQ5D, synovitis, or VAS for this donor to avoid introducing bias based on inflammatory status or clinical scores that could inadvertently influence downstream interpretations.”

Comments 4: line 145-150: Why was the cut-off of 21 used while it is often seen that the cut-of is 20? Similar why were these specific cut-offs chosen for the EQ5D. What was the amount of patients not fitting one of these groups.

Resposne 4: Thank you for this important question. The cut-offs for OKS (≤21 for severe, ≥22 for mild/moderate) and EQ5D (≥9 for severe, ≤8 for mild/moderate) were selected to achieve a clear stratification of patients into two distinct groups while maintaining approximately balanced group sizes. Although a cut-off of OKS ≤20 is sometimes used in the literature, we selected 21 as a pragmatic threshold based on the distribution of our cohort data and to allow for a small buffer zone between groups. EQ5D cut-offs were chosen using a similar rationale, as patient-reported outcomes can be variable and often benefit from broader binning to reduce misclassification noise.

This approach enabled us to identify a middle “unclassified” subset of patients who did not clearly fall into either group, which we considered appropriate given the subjective nature of the scoring tools. In total, n = 71 patients were analysed:

  • n = 29 classified as severe OA
  • n = 24 classified as mild/moderate OA
  • n = 18 did not meet the criteria for either group

These details have now been clarified in the Results section.
Line 145: “Patients were stratified into mild/moderate and severe OA groups based on thresholds of OKS ≤21 and EQ5D ≥9 (severe), or OKS ≥22 and EQ5D ≤8 (mild/moderate), selected to create distinct groups based on the data distribution and allow a buffer between classifications. Of 71 patients used for this analysis, 29 were classified as severe OA, 24 were classified as mild/moderate OA and 18 did not meet the criteria for either group.”

Comments 5: As there is no control how specific these findings are (yes there is difference between mild vs severe based on these criteria) it would be helpful to see if the patients not fulfilling these criteria have also a different SFEVprofile, please discuss. 

Response 5: Thank you for this suggestion. We agree that including patients outside of the defined mild/moderate and severe OA categories could provide additional insight into the continuum of SFEV characteristics. However, our primary goal was to compare two clearly stratified groups to minimise ambiguity introduced by subjective patient-reported outcome measures. Including an intermediate group risked increasing heterogeneity and reducing statistical power.

That said, this is an interesting point and warrants further exploration in future studies. It is plausible that patients not meeting our strict group definitions (n = 18) may exhibit intermediate SFEV profiles or greater variability, which could reflect the continuous nature of OA severity. We have now acknowledged the potential value of further subgroup analyses in the Discussion section.

Discussion: “Although patients were stratified into discrete groups for clarity, OA exists along a continuum. Future analyses could explore whether patients who did not meet our severity thresholds exhibit intermediate or distinct SFEV characteristics.”

Comments 6: Results:

I don't understand the n-values presented in table 1 or specific experimentsts. Why are the data of only 7 mild patients and 4 severe patients presented while in line 172 13 patients (mild) are mentioned.  It is remarkable that the age of the severe patients is relatively young. This suggests that this is a fast progressive group of patients which may interfere with the profile you suggest. Why is an asterisk presented in the table? To highlight  the significance of OKS difference?  Please clarify this in the text 

Response 6: Thank you for raising these points. We appreciate the opportunity to clarify.

Thank you for your observation. We have now corrected this error in the manuscript. “EVs were isolated by ultracentrifugation from the synovial fluid of n = 13 OA patients and characterised by NTA (Figure 1B). These included patients with mild/moderate OA (n = 7), severe OA (n = 4), and two patients who did not meet the classification criteria.”

In some cases, patient-reported outcome data (e.g., EQ5D or OKS) were incomplete or missing, which explains minor discrepancies in group totals across different analyses. We have clarified this in the figure and table legends.

Regarding the comment on age, our statistical analysis shows there was no difference between mild/moderate and severe groups (p = 0.5037) and therefore we do not believe age to be a confounding variable in this cohort. The asterisk in Table 1 is intended to denote a statistically significant difference in OKS between mild/moderate and severe groups (p = 0.0443). This has now been explicitly defined in the table legend.

We have updated the manuscript accordingly:

  • Methods section: Clarifies the rationale and constraints for selecting the EV analysis subset. “A subset of 11 patients were selected for SFEV characterisation. These consisted of 7 patients classified as mild/moderate OA and 4 classified as severe OA based on EQ5D and OKS scores. In some cases, individual patient-reported outcome measures were incomplete, resulting in minor discrepancies in n-values across different analyses (Table 1.)”
  • Table 1 legend: Now includes “*p < 0.05, significant difference between groups” to explain the asterisk.
  • Discussion section: “OKS scores were significantly different between the two groups (p = 0.0443), supporting a distinction in patient-reported disease severity.”

Comments 7: Line 196: does it matter which amount of EVs or synovial fluid is added (is there data available) and why is there a range used for synovial fluid. On what data is this based?

Response 7: Thank you for your comment and for the opportunity to clarify. We have adjusted this appropriately in the revised Methods section.

Chondrocytes were treated with either isolated SFEVs (5 × 10⁹ particles per well, 48 well plate) or with synovial fluid volumes predicted to contain an equivalent number of EVs, based on measured isolation efficiency. This corresponded to 4.5μl per well for the severe OA group and 8μl per well for the mild/moderate OA group. These volumes represented a small fraction of the total culture volume and were kept consistent within each group.

Comments 8: discussion: line 479-480 although indeed neovascularization is demonstrated in more severe patients this is not as black and white as stated here. Especially as this is not assessed in this study, this section feels a bit speculative.

Response 8: Thank you for your feedback. We accept the reviewer’s comments that the role and importance of neovascularisation in OA pathology is not fully understood, and that we did not assess neovascularisation in our study. Our intention to refer to neovascularisation in the discussion was simply to acknowledge that it has previously been reported as a pathological feature in OA progression, and thus it provided some context to our findings on ICAM and ANGPTL7 without overstating the functional implications. In the revised manuscript we rephrased this, and added additional references  Bonnet & Walsh 200 (doi.org/10.1093/rheumatology/keh344) and Walsh et al., 2010 (10.1093/rheumatology/keq188) to the discussion on the evidence of neovascularisation in OA pathology to better acknowledge the research field in this area.    

Comments 9: General remarks: In the introduction and discussion the emphasis is on the inflammatory OA patient; what makes the donors used for EV isolation be these type of patients?

Response 9: Thank you for your comment. All patients recruited to this study exhibited significant synovitis as observed by MRI. Scoring synovitis across 11 compartments across the knee joint, 11 patients were classified as having severe synovitis and 2 patients were classified as having moderate synovitis. Based on the severity of the synovitis across multiple compartments of the joint, our patients likely represent an inflammatory OA endotype.  Our recruitment of patients to the study excluded those who have secondary OA due to a traumatic joint injury, and this exclusion criteria may have biased our patients towards those with a more inflammatory endotype.

In the revised manuscript, we have updated Table 1 patient characteristics with the mean synovitis score. Furthermore, we have referred to the exclusion criteria in the methods section, and have reflected on this point in the discussion. “Although patients were not selected based on inflammatory biomarkers, the cohort studied likely represents an inflammatory OA endotype- all patients exhibited synovitis by MRI, assessed across 11 compartments of the knee, and all patients were classified as having moderate or severe synovitis. Importantly, synovitis scores were not significantly different between the mild/moderate and severe OA groups, indicating that inflammation was not a driving factor in the stratification but was a common feature across the cohort. This inflammatory background may help explain the strong pro-inflammatory responses observed in chondrocytes treated with SFEVs.”

Comments 10: Why was histology not considered as assessment of cartilage severity and/or neovascularization? Similar for synovial tissue histology which might provide data on inflammatory status.

Response 10: Thank you for your comment. While we agree that histological analysis of cartilage and synovium can provide valuable insight into tissue-level disease processes, this was not feasible within the scope of the current study. Cartilage tissue was fully allocated for primary chondrocyte isolation. Additionally, our study was specifically focused on the characterisation of synovial fluid extracellular vesicles and their functional effects on chondrocytes, rather than on synovial tissue pathology per se.

We acknowledge that incorporating histological data would provide a complementary view of inflammatory status and vascular changes, and we highlight this as an important avenue for future studies. “Future studies incorporating histological analysis of cartilage and synovium could further validate EV-associated inflammatory or vascular signatures and help refine OA endotype classification.”

Reviewer 2 Report

Comments and Suggestions for Authors

The work deals with a very interesting and current topic, namely extracellular vesicles and their role in disease and I read with interest this study. The work is beneficial in this unexplored area. Here are my questions and comments:

Results 3.3:

…To determine the functional effect of SFEVs, articular chondrocytes were cultured in the presence of either SFEVs (5x109), or synovial fluid (4-8μl/ml culture media) containing the equivalent amount of EVs…

You use the cell culture of monolayer chondrocytes cultures at which passage? Without a biomechanical environment? What were the phenotypic markers of the cell passage used? Were these chondrocytes already dedifferentiating producing type I collagen, and do they still produce type II collagen as well as aggrecan which are distinct chondrocyte biomarkers?

I assume that the EVs or synovial fluid were diluted with the chondrocyte growth medium before adding to the chondrocytes. How did you address the possible additive effect of EVs from FCS in the medium on the chondrocytes themselves?

Author Response

Comments 1: Results 3.3: You use the cell culture of monolayer chondrocytes cultures at which passage? Without a biomechanical environment? What were the phenotypic markers of the cell passage used? Were these chondrocytes already dedifferentiating producing type I collagen, and do they still produce type II collagen as well as aggrecan which are distinct chondrocyte biomarkers?

Response 1: Thank you for this detailed question. Chondrocytes were cultured in standard monolayer and used at early passage (P2) to minimise any potential phenotypic drift. While we did not assess COL1A1, COL2A1 or ACAN expression directly in this study, we selected P2 based on previous reports showing that chondrocytes retain expression of key phenotypic markers, including type II collagen and aggrecan, at this passage under similar culture conditions (Dell’Accio et al., Arthritis and Rheumatology, 2001). We acknowledge that dedifferentiation remains a limitation of 2D culture, and that incorporation of biomechanical stimuli or 3D systems could further preserve native chondrocyte phenotype — an important consideration for future studies. The manuscript has been updated to reflect these details “Cartilage was obtained from macroscopically intact regions of the joint from individual patients undergoing total knee arthroplasty. As anatomical compartmentalisation was not recorded, samples likely reflect a mixture of femoral, tibial, or patellar articular cartilage. Fresh cartilage was dissected into 1-3mm3 pieces, and digested using 2mg/ml collagenase (Merck, C9891) in DMEM (Merck, D6429) for 7 hours at 37˚C. Chondrocytes were cultured in standard monolayer and used at early passage (P2) to minimise dedifferentiation. The same donor-derived chondrocyte population (BMI: 28.36) was used across all treatment conditions to minimise inter-donor variability. No data were collected on OKS, EQ5D, synovitis, or VAS for this donor to avoid introducing bias based on inflammatory status or clinical scores that could inadvertently influence downstream interpretations.”. “Chondrocytes were treated with either pooled SFEVs or pooled synovial fluid from patients with mild/moderate or severe OA. Pooled samples were created by combining equal volumes from four classified donors per group (n = 4 mild/moderate, n = 4 severe). Treatments were carried out in serum-free media for 24 h to prevent interference from serum-derived EVs.”

Comments 2: I assume that the EVs or synovial fluid were diluted with the chondrocyte growth medium before adding to the chondrocytes. How did you address the possible additive effect of EVs from FCS in the medium on the chondrocytes themselves?

Response 2: Thank you for highlighting this important point. To avoid confounding effects from extracellular vesicles present FCS, chondrocytes were cultured in serum-free media during the 24-hour treatment period with either SFEVs or synovial fluid. This ensured that any observed effects were attributable to the added SFEVs or synovial fluid rather than to EVs derived from serum supplements.

We have now added this clarification to the Methods section.

“All treatments were performed in serum-free media for 24 hours to prevent interference from serum-derived EVs.”

Reviewer 3 Report

Comments and Suggestions for Authors

Introduction:

The phrase starting with “Classified by their size…” is unclear as exosomes, microvesicles and apoptotic bodies are not differentiated by size but rather by biogenesis; also the given citation does not confirm any such classification.

The introduction emphazises the importance of OA endotypes and focuses on the inflammatory endotype. However, the relevance of the study findings for explaining or targeting the inflammatory endotype remains unclear.

Methods:

Useful information on the procedure of SF collection, the yield (ml) of SF, the kind of surgeries performed are missing. Did the authors undertake any measures to assess the presence or absence of an inflammatory endotype?

Where does the big range in cartilage digestion time (5-15h) stem from?

Were the chondrocytes passaged? To which passage? If higher passage numbers were used, did the authors confirm the chondrocyte phenotype (and how)?

How much SF was used for EV isolation, and which amount of EVs was yielded?

100,000 xg for 16h is usually used for EV depletion – also in the reference cited. For isolation, shorter times are used. What was the reason for this long duration of centrifugation?

Line 103: Please specify the optimal dilution.

Were the EVs from the 13 patients pooled? Were the pooled EVs used for characterization?

The method for preparing the SFEV lysate should be described.

Explain the rational for using a cutoff of 21 to separate groups according to the OKS.

A major concern with the work arises from the fact that the discrimination between severe and mild OA is solely based on questionnaires. Despite the undoubted value of self reported outcome measures, other parameters such as imaging or molecular biomarkers could provide a better explanation for the observed differences between the EVs of the 2 groups. As it is now, the marked differences between the effects of the EVs from “severe” and “mild/moderate” that are solely based on questionnaires is astonishing and not explainable biologically or clinically.

Results:

Throughout the results it remains unclear whether the SF samples were the supernatants resulting from EV depletion/isolation or the full SF comprising all EVs.

The labelling of x- and y-axis of the figure panels inofmost figures, as well as the labelling of the cartoons (e.g. Figure 1B) are too small.

Figure 1C: What is being shown? Dilution? Severe/mild?

Figure 1F: Dilution? Severe/mild? What are the numbers in the images?

Figure 1G: Explain the groups on the x-axis. It remains unclear what is being shown. Is the difference a result of different EV amounts between the two groups or is it really a difference in (the extent of) tetraspannin expression? In fact, the pattern is rather similar. Further experimental details (normalization?) of the samples loaded to the ExoView might be necessary.

The patient characteristics (Table 1) presented in section 3.2. should be shilfted to section 3.1.

Line 171/172: ultracentrifugation was only perfomed for mild/moderate samples?

The patient numbers are unclear and do not match between Table 1, section 3.2. and section 3.1.

Line 172: In fact, NTA is not a method to confirm EVs, but of particles including EVs.

Line 176: Does a correlation of EV concentration with EQ5D makes sense? What does it mean?

Line 197: For treatment of chondrocytes, SF was adjusted to contain the equivalent amount of EVs compared to SFEV preparations. However, the methods of analyzing the EV content in SF was not described.

Legend Figure 2: The amount/concentration of EVs used for treatment should be specified.

Supplementary Tables 1 and 2: Why is column C the same in both tables?

Line 227-254: Both treatment groups were compared to the untreated group. Why not also compared both treatments to each other? Same for Line 330ff.

Where does the cutoff log2FC>0.58 stem from?

Figure 4: panel A does not illustrate what is described in results and figure legend. Although interesting, the data shown in Figure 4B and C do not provide an explanation for the observed differential effects of SFEVsevere and SFEVmild. Other factors appear to be responsible for this differential effect. Experiments that assess these factors would be of great benefit for the impact of the work. At present, the data in Figure 4 represent a characterization of the respective EVs, but should be presented earlier in the manuscript.

Line 435: It is unclear whether the SFEV lysates were prepared from EVs from patients with severe or mild/moderate OA.

Discussion:

Again, the role of synovial inflammation in OA is highlighted, but the association of this fact to the present study is unclear. “Severe” OA, as assessed via questionnaires is not necessarily related to synovial inflammation. This issue is a major concern with the present work. As such, no difference in IL6 between EVs from severe and mild/moderate OA was found. Thus, inflammatory factors might not be the relevant agents that explains the different impact.

Line 539-542: This is unclear, as – in fact – the authors should have an EV-depleted SF control in their hands, as they ultracentrifuged the SF for 16h, which by definition us a method for EV depletion. The resulting supernatant would represent such a EV-depleted SF control.

Author Response

Comments 1: Introduction: The phrase starting with “Classified by their size…” is unclear as exosomes, microvesicles and apoptotic bodies are not differentiated by size but rather by biogenesis; also the given citation does not confirm any such classification.

Response 1: Thank you for highlighting this. We agree that the original phrasing was imprecise and may conflate classification by size with classification by biogenesis. To improve clarity, we have revised the text in the Introduction to distinguish between these two approaches to EV categorisation.

“EVs are broadly heterogeneous and can be classified by both size and biogenesis. Small EVs (typically <200 nm) and larger EVs (>200 nm) encompass populations such as exosomes (originating from endosomal multivesicular bodies) and microvesicles (formed by outward budding of the plasma membrane), respectively.”

Comments 2: The introduction emphasises the importance of OA endotypes and focuses on the inflammatory endotype. However, the relevance of the study findings for explaining or targeting the inflammatory endotype remains unclear.

Response 2: Thank you for this comment. We agree that the link between our findings and the inflammatory OA endotype could be more explicitly stated. Although we did not pre-select patients based on inflammatory biomarkers, all patients recruited for this study exhibited significant synovitis as determined by scoring of synovitis across 11 joint compartments by MRI. Indeed, 11 patients were classified as having severe synovitis, and the remaining 2 patients classified as having moderate synovitis. Furthermore, our severe OA group, stratified by OKS and EQ5D, exhibited elevated tetraspanin expression on SFEVs and a consistent induction of inflammatory mediators (e.g., IL-6, CXCL5, ICAM1) and nociceptive markers (e.g., NGF, Substance P) in treated chondrocytes. These findings support the concept that this subgroup likely reflects an inflammatory OA endotype at the molecular level.

We have now updated Table 1 to include the synovitis scores and clarified these points in the Discussion section, highlighting that the EV-driven pro-inflammatory and pro-nociceptive responses observed in our study are consistent with features of the inflammatory endotype described in the OA literature. “Although patients were not selected based on inflammatory biomarkers, the cohort studied likely represents an inflammatory OA endotype- all patients exhibited synovitis by MRI, assessed across 11 compartments of the knee, and all patients were classified as having moderate or severe synovitis. Importantly, synovitis scores were not significantly different between the mild/moderate and severe OA groups, indicating that inflammation was not a driving factor in the stratification but was a common feature across the cohort. This inflammatory background may help explain the strong pro-inflammatory responses observed in chondrocytes treated with SFEVs.”, “These findings suggest that the severe OA group in our study likely reflects an inflammatory endotype, as evidenced by the SFEV-induced upregulation of pro-inflammatory (e.g., IL-6, CXCL5, ICAM1) and nociceptive (e.g., NGF, TAC1) mediators in chondrocytes.”

Comments 3: Methods: Useful information on the procedure of SF collection, the yield (ml) of SF, the kind of surgeries performed are missing. Did the authors undertake any measures to assess the presence or absence of an inflammatory endotype?

Response 3: Thank you for your comment. We have clarified the details of synovial fluid collection in the Methods section. Briefly, synovial fluid was collected intra-operatively from patients undergoing total knee replacement surgery for osteoarthritis, immediately after joint capsule opening. Collection volumes varied across patients, ranging from approximately 100 μL to 8 mL, depending on availability.

We did not pre-select patients based on inflammatory markers. As mentioned above, all patients exhibited significant synovitis across multiple compartments of the knee joint. We then used stratification based on Oxford Knee Score (OKS) and EQ5D to define mild/moderate and severe OA groups. These metrics, while not direct measures of synovial inflammation, have previously been associated with symptom burden and functional impairment, which are often linked to inflammatory OA endotypes. Furthermore, SFEVs from the severe group induced elevated expression of pro-inflammatory and nociceptive mediators in chondrocytes (e.g., IL-6, CXCL5, ICAM1, NGF), suggesting that this subgroup reflects an inflammatory OA phenotype at the molecular level. This point is now more clearly discussed in the revised Discussion section.

Comments 4: Where does the big range in cartilage digestion time (5-15h) stem from?

Response 4: Thank you for your spotting this error. Cartilage digestion was monitored continuously until the cartilage fragments were visibly digested and a uniform single-cell suspension was achieved. This digestion process took approximately 7 hours. We have updated the Methods section accordingly.

“Fresh cartilage was dissected into 1-3mm3 pieces, and digested using 2mg/ml collagenase (Merck, C9891) in DMEM (Merck, D6429) for 7 hours at 37˚C.”

Comments 5: Were the chondrocytes passaged? To which passage? If higher passage numbers were used, did the authors confirm the chondrocyte phenotype (and how)?

Response 5: Thank you for your comment. Briefly, chondrocytes were used at early passage (P2) to minimise dedifferentiation. While we did not assess marker expression directly, we selected this passage based on published data showing retention of key phenotypic markers at P2 under similar conditions. This has now been clarified in the methods section. “Cartilage was obtained from macroscopically intact regions of the joint from individual patients undergoing total knee arthroplasty. As anatomical compartmentalisation was not recorded, samples likely reflect a mixture of femoral, tibial, or patellar articular cartilage. Fresh cartilage was dissected into 1-3mm3 pieces, and digested using 2mg/ml collagenase (Merck, C9891) in DMEM (Merck, D6429) for 7 hours at 37˚C. Chondrocytes were cultured in standard monolayer and used at early passage (P2) to minimise dedifferentiation.

Comments 6: How much SF was used for EV isolation, and which amount of EVs was yielded?

Response 6: Thank you for your comment. For each patient, 1 mL of synovial fluid was used for EV isolation. Following ultracentrifugation, the EV pellet was resuspended in 100 μL of PBS, corresponding to 10% of the original volume. Nanoparticle tracking analysis (NTA) data, shown in Figure 1C, reflects the range of EV concentrations obtained across individual samples.

We have now clarified these details in the Methods section. “For each patient, 1 mL of synovial fluid was used for EV isolation. Following ultracentrifugation, the EV pellet was resuspended in 100 μL of PBS, corresponding to 10% of the original volume.”

Comments 7: 100,000 xg for 16h is usually used for EV depletion – also in the reference cited. For isolation, shorter times are used. What was the reason for this long duration of centrifugation?

Response 7: Thank you for this important comment. While longer ultracentrifugation spins such as 100,000 ×g for 16 hours are typically associated with EV depletion protocols, there is clear evidence that such conditions also pellet nanoparticulate material, including small EVs (sEVs). Zhang et al. (https://doi.org/10.1016/j.celrep.2019.01.009) demonstrated that a 16-hour ultracentrifugation following standard EV isolation still yields a pellet containing discrete extracellular particles (termed exomeres), emphasizing that extended centrifugation can actively recover EV subpopulations rather than merely clearing debris. In the absence of a initial 2-4hr centrifugation we are likely to be maximising the yield of conventional sEV and potentially exomeres too- but without clear evidence of their biogenesis pathway we prefer to stick with the MISEV term of sEV.

In our study, the choice of a 16-hour spin was additionally motivated by the properties of the starting material. Synovial fluid remains relatively viscous even after hyaluronidase treatment, and shorter spins may be insufficient to efficiently sediment small vesicles under such conditions. A longer duration was thus necessary to maximize EV recovery and ensure a representative isolation.

We are aware that extended ultracentrifugation could potentially promote particle aggregation. However, given the need to overcome viscosity-related sedimentation limitations, we judged that maximizing recovery was essential. To monitor for potential aggregation, we performed Nanoparticle Tracking Analysis (NTA) and observed no evidence of abnormal size distributions or aggregation artifacts in our isolated vesicles.

Comments 8: Line 103: Please specify the optimal dilution.

Response 8: Thank you for highlighting this. We have now updated the methods to reflect the 1:200 dilution used for NTA characterisation. “The optimal dilution (1:200) was determined by...”

Comments 9: Were the EVs from the 13 patients pooled? Were the pooled EVs used for characterization?

Response 9: Thank you for your question. No, EVs were not pooled for characterization. Nanoparticle tracking analysis (NTA), ExoView, and protein cargo profiling were all performed on individual patient-derived EV samples. Pooled samples were used only for downstream chondrocyte treatment experiments, where EVs or synovial fluid from 4 donors per group (mild/moderate or severe) were combined in equal volumes to reduce inter-donor variability and ensure sufficient material. We have clarified this distinction in the revised Methods section.

Comments 10: The method for preparing the SFEV lysate should be described.

Response 10: Thank you for your comment. The method for preparing the SFEV lysates is described in the Methods section 2.5: “prior to protein quantification, hyaluronidase-treated synovial fluid samples and isolated SFEVs were lysed with 0.5% Triton X-100, and diluted 1:2 with dilution buffer.”

Comments 11: Explain the rational for using a cutoff of 21 to separate groups according to the OKS.

Response 11: Thank you for your comment. The cut-offs for OKS (≤21 for severe, ≥22 for mild/moderate) and EQ5D (≥9 for severe, ≤8 for mild/moderate) were selected to achieve a clear stratification of patients into two distinct groups while maintaining approximately balanced group sizes. Although a cut-off of OKS ≤20 is sometimes used in the literature, we selected 21 as a pragmatic threshold based on the distribution of our cohort data and to allow for a small buffer zone between groups. EQ5D cut-offs were chosen using a similar rationale, as patient-reported outcomes can be variable and often benefit from broader binning to reduce misclassification noise.

This approach enabled us to identify a middle “unclassified” subset of patients who did not clearly fall into either group, which we considered appropriate given the subjective nature of the scoring tools. In total, n = 71 patients were analysed:

  • n = 29 classified as severe OA
  • n = 24 classified as mild/moderate OA
  • n = 18 did not meet the criteria for either group

These details have now been clarified in the Results section.
Line 145: “Patients were stratified into mild/moderate and severe OA groups based on thresholds of OKS ≤21 and EQ5D ≥9 (severe), or OKS ≥22 and EQ5D ≤8 (mild/moderate), selected to create distinct groups based on the data distribution and allow a buffer between classifications. Of 71 patients used for this analysis, 29 were classified as severe OA, 24 were classified as mild/moderate OA and 18 did not meet the criteria for either group.”

Comments 12: A major concern with the work arises from the fact that the discrimination between severe and mild OA is solely based on questionnaires. Despite the undoubted value of self reported outcome measures, other parameters such as imaging or molecular biomarkers could provide a better explanation for the observed differences between the EVs of the 2 groups. As it is now, the marked differences between the effects of the EVs from “severe” and “mild/moderate” that are solely based on questionnaires is astonishing and not explainable biologically or clinically.

Response 12: Thank you for this comment. We don’t agree that stratifying our patients based on patient-reported outcome measures (PROMs) is a major concern. The PROMs used in our study to stratify patients i.e Oxford Knee Score (OKS) and EQ5D are well-validated, widely used clinical tools that reflect joint function, pain, and overall quality of life — key outcomes from the patient’s perspective, and are therefore informative in guiding the clinical decision for joint replacement surgery [Conaghan et al. doi.org/10.1002/art.23091 , Bone Joint J 2019;101-B(6 Supple B):23–30], as well as in determining post-operative outcomes for patients (Wailoo et al., doi: 10.1186/1477-7525-12-37).

While there can be a disconnect between structural joint pathology (e.g., radiographic scores or imaging) and symptom burden, PROMs capture functional and pain-related dimensions that imaging or molecular biomarkers may fail to reflect. Indeed, studies have shown that structural severity does not always correlate with clinical symptoms in OA [Hunter, DJ et al., Osteoarthritis and Cartilage, 2013, doi.org/10.1016/j.joca.2013.05.017].

Nonetheless, we agree with the reviewer that incorporating additional objective measures — such as imaging data or molecular biomarker profiling — could further validate and refine patient stratification. We have now included the following in the revised Discussion section to acknowledge this point:

“It should also be noted that we categorised OA severity based on patient-reported outcomes, including the well-established Oxford Knee Score [doi.org/10.1002/art.23091 , DOI: 10.1302/0301-620X.101B6.BJJ-2018-1460.R1] and EQ5D (doi: 10.1186/1477-7525-12-37). These instruments collectively assess joint mobility, function, pain, and quality of life — aspects that are clinically meaningful and directly relevant to patients. However, given the partial disconnect between patient-reported outcome measures and structural pathology in OA [doi.org/10.1016/j.joca.2013.05.017], future studies applying complementary stratification methods, such as MRI-based assessments or molecular biomarker profiling, would help to further validate EV-associated phenotypes and disease severity classifications.”

Comments 13: Results: Throughout the results it remains unclear whether the SF samples were the supernatants resulting from EV depletion/isolation or the full SF comprising all EVs.

Response 13: Thank you for your comment. We confirm that no EV-depleted supernatants were used in this study. All synovial fluid used for chondrocyte treatment was untreated whole synovial fluid, and not processed through ultracentrifugation prior to use. Although we initially intended to include an EV-depleted synovial fluid control, this proved technically challenging due to the high viscosity of synovial fluid — even after hyaluronidase treatment. Multiple rounds of ultracentrifugation failed to reliably deplete EVs without extensive dilution of the sample. Importantly, any alternative depletion method (e.g. size exclusion chromatography, filtration, or chemical precipitation) would have altered the composition of the synovial fluid itself, introducing additional variables and requiring separate controls. For this reason, we elected to use full synovial fluid (with EVs intact) to preserve physiological relevance and comparability with the EV-only condition. Our original discussion of this has now been further expanded to address this “Although inclusion of an EV-depleted synovial fluid control would have strengthened this comparison, reliable depletion was not feasible due to the high viscosity of synovial fluid. Methods that could remove EVs (e.g. repeated ultracentrifugation or size exclusion) would have altered the composition of the fluid, necessitating additional control groups and complicating interpretation. Therefore, full synovial fluid was used to maintain comparability and physiological relevance.”

Comments 14: The labelling of x- and y-axis of the figure panels inofmost figures, as well as the labelling of the cartoons (e.g. Figure 1B) are too small.

Response 14: In the revised manuscript we have increased the labelling of axis and text in all figure panels.

Comments 15: Figure 1C: What is being shown? Dilution? Severe/mild?

Response 15: Thank you for your question. Figure 1C shows a combined Nanoparticle Tracking Analysis (NTA) trace illustrating the size distribution profile of synovial fluid EVs isolated from all 13 patients. We have now clarified this in the figure legend. “Histogram showing the average and size distribution of synovial fluid extracellular vesicles (SFEVs) isolated by centrifugation, as measured by NTA (n = 13 patients, n=7 mild/moderate OA, n=4 severe OA, n=2 excluded from either group).”

Comments 16: Figure 1F: Dilution? Severe/mild? What are the numbers in the images?

Response 16: Thank you for your question. Figure 1F shows a representative ExoView image of synovial fluid EVs stained for CD9, CD63, and CD81 tetraspanins. The image was acquired from a mild/moderate OA patient sample, using a 1:500 dilution of the EV preparation. We have now clarified the dilution in both the Methods section and the figure legend. The numbers shown in the image correspond to internal software-generated identifiers and are not relevant to data interpretation. “(F) Representative ExoView fluorescence image of SFEVs from a mild/moderate OA patient (dilution 1:500). Tetraspanins shown: CD9 (blue), CD63 (red), CD81 (green).”, “EVs were further characterised using the ExoView R100 reader (Unchained Labs, Malvern, United Kingdom) to analyse size, concentration, and CD9, CD63 and CD81 tetraspanin markers using the Leprechaun Exosome Human Plasma Kits (251-1045, Unchained Labs, Malvern, United Kingdom) at a dilution of 1:500, as previously described [16].

Comments 17: Figure 1G: Explain the groups on the x-axis. It remains unclear what is being shown. Is the difference a result of different EV amounts between the two groups or is it really a difference in (the extent of) tetraspanin expression? In fact, the pattern is rather similar. Further experimental details (normalization?) of the samples loaded to the ExoView might be necessary.

Response 17: Thank you very much for your observations here. We agree that the x-axis in Figure 1G was not sufficiently explained. The labels represent tetraspanin capture antibodies (CD9, CD63, CD81) immobilised on the ExoView chip, followed by staining with corresponding fluorescently-labelled detection antibodies, allowing quantification of EVs expressing each tetraspanin. The figure legend has been updated to reflect this. “(G) Two-way ANOVA analysis of tetraspanin expression on SFEVs from severe OA and mild/moderate OA patients, as measured by ExoView (p = 0.004, n=13). Tetraspanin-positive EV counts detected on ExoView chips functionalised with CD9, CD63, or CD81 capture antibodies and stained with corresponding fluorescent detection antibodies. EVs from mild/moderate and severe OA patients were analysed at 1:500 dilution. Despite lower EV concentrations in the severe OA group (as shown in Figure 1D), ExoView revealed a higher number of tetraspanin-positive EVs in this group, suggesting enrichment of tetraspanin expression.”

Importantly, while EV concentration (as measured by NTA) was lower in patients with worse quality of life (i.e. higher EQ5D scores), we observed a higher tetraspanin-positive particle count in the same group via ExoView. This indicates that, despite lower overall EV numbers, the vesicles present in patients with more severe symptoms were enriched in surface tetraspanin expression. This relationship was incorrectly stated in the manuscript and has now been corrected. “A significant negative correlation was observed between EQ5D score and SFEV concentration (p = 0.0067, r² = 0.5025, Figure 1D), indicating that lower quality of life was associated with lower SFEV concentrations.”, “Figure 1E) suggesting that patients with worse quality of life tended to have larger SFEVs.”

Comments 18: The patient characteristics (Table 1) presented in section 3.2. should be shilfted to section 3.1.

Response 18: Thank you. We have moved Table 1 to section 3.1.

Comments 19: Line 171/172: ultracentrifugation was only perfomed for mild/moderate samples?

Response 19: Thank you for your observation. We have now corrected this error in the manuscript. “Particles isolated by ultracentrifugation from the synovial fluid of n = 13 OA patients were analysed by Nanoparticle Tracking Analysis (NTA) to determine size distribution and concentration. These included patients with mild/moderate OA (n = 7), severe OA (n = 4), and two patients who did not meet the classification criteria.”

Comments 20: The patient numbers are unclear and do not match between Table 1, section 3.2. and section 3.1.

Response 20: Thank you for this observation. The differences in patient numbers reflect the use of distinct subsets of the full patient cohort for different analyses, which we have now clarified in the manuscript.

  • n = 71 patients completed OKS and EQ5D scoring and were included in the initial stratification analysis (Section 3.1; Figure 1A).
  • n = 13 patients were used for EV isolation and full characterisation via NTA, ExoView, and protein cargo analysis (Section 3.2; Figure 1C–G).
  • Of these 13, n = 11 patients (7 mild/moderate, 4 severe) were included in group-based analyses as two patients were excluded from both groups in stratification.

We have updated the relevant sections and figure legends to clarify this and improve consistency across the manuscript.

Comments 21: Line 172: In fact, NTA is not a method to confirm EVs, but of particles including EVs.

Response 21: Thank you for pointing this out. We agree that Nanoparticle Tracking Analysis (NTA) does not specifically identify extracellular vesicles, but rather quantifies particles within a defined size range, which may include EVs alongside other co-isolated components. We have clarified this in the manuscript to ensure accurate terminology. “Particles isolated by ultracentrifugation from the synovial fluid of n = 13 OA patients were analysed by Nanoparticle Tracking Analysis (NTA) to determine size distribution and concentration.”

Comments 22: Line 176: Does a correlation of EV concentration with EQ5D makes sense? What does it mean?

Response 22: Thank you for this insightful question. EQ5D is a validated measure of overall health-related quality of life, incorporating dimensions such as mobility, pain, and daily function — all of which are commonly impaired in advanced OA. The observed negative correlation between SFEV concentration and EQ5D suggests that patients reporting worse quality of life (i.e., higher EQ5D scores) tended to have lower EV concentrations in their synovial fluid.

While we acknowledge that the link between EV concentration and subjective clinical measures should be interpreted with caution, this correlation may reflect disease-associated alterations in joint cellular activity, synovial fluid composition, or vesicle turnover. It is also possible that more severe joint dysfunction is associated with reduced EV release due to synovial tissue degradation, fibrosis, or altered cell viability. However, this remains hypothetical, as the current study was not designed to investigate the underlying mechanisms driving changes in EV abundance. We have clarified this point in the Discussion section. “We have been able to categorise patients into mild/moderate and severe OA groups using patient-reported quality of life, pain and joint function combined data. The inverse relationship between EV concentration and EQ5D suggests that lower EV abundance may be associated with more advanced disease or reduced joint function. One possible explanation is that synovial tissue in severe OA becomes fibrotic, inflamed, or otherwise dysfunctional, leading to altered EV production or release. However, this remains speculative, as our study did not directly assess tissue pathology or EV biogenesis mechanisms.”

Comments 23: Line 197: For treatment of chondrocytes, SF was adjusted to contain the equivalent amount of EVs compared to SFEV preparations. However, the methods of analyzing the EV content in SF was not described.

Response 23: Thank you for this observation. EV concentration in the pooled, raw (hyaluronidase-treated) synovial fluid was first quantified using Nanoparticle Tracking Analysis (NTA) prior to ultracentrifugation. These measurements were then used to calculate the volume of synovial fluid required to match the EV number used in SFEV treatments (5 × 10⁹ particles per well). This corresponded to 4.5 μL of synovial fluid per well for the severe OA group and 8 μL for the mild/moderate group, based on the percentage isolation efficiency (1.5% and 2.66%, respectively). These volumes were kept consistent within groups for all chondrocyte treatment experiments. We have clarified this point in the revised Methods section. “Chondrocytes were treated with either isolated SFEVs (5 × 10⁹ particles per well) or with synovial fluid volumes calculated to contain an equivalent number of EVs. To determine these volumes, pooled, hyaluronidase-treated synovial fluid was analysed by Nanoparticle Tracking Analysis (NTA) prior to ultracentrifugation. Based on the measured EV concentrations and the calculated EV isolation efficiency for each group (1.5% for severe OA and 2.66% for mild/moderate OA), we estimated the volume of whole synovial fluid required to yield 5 × 10⁹ EVs. This corresponded to 4.5 μL per well for severe OA samples and 8 μL per well for mild/moderate OA samples. These volumes represented a small fraction of the total culture media and were kept consistent within each group for all experiments. All treatments were performed in serum-free media for 24 hours to prevent interference from serum-derived EVs.”

Comments 24: Legend Figure 2: The amount/concentration of EVs used for treatment should be specified.

Response 24: Thank you for noticing this. We have now amended the figure legend. “Chondrocytes were treated with either 5 × 10⁹ SFEV particles per well (48 well plate) or with synovial fluid volumes containing an equivalent number of EVs: 4.5 μL per well for severe OA and 8 μL per well for mild/moderate OA, based on measured EV concentration and isolation efficiency.”

Comments 25: Supplementary Tables 1 and 2: Why is column C the same in both tables?

Response 25: Column C displays the average expression value for each gene across all treatment groups in the study, and is therefore the same for both supplementary Table 1 and 2.

Comments 26: Line 227-254: Both treatment groups were compared to the untreated group. Why not also compared both treatments to each other? Same for Line 330.

Response 26: We thank the reviewer for this suggestion. Our primary objective was to evaluate how synovial fluid EVs (SFEVs) from different OA severities modulate chondrocyte gene expression relative to an untreated baseline. This approach allowed us to identify both common and severity-associated transcriptional responses, using consistent thresholds (log2FC ≥ 0.58 and FDR ≤ 0.05) applied to each comparison.

We opted not to emphasize a direct comparison between severe and mild/moderate OA SFEV-treated groups because the DEGs identified as “unique to severe OA” were defined relative to untreated cells—not because they were statistically different from the mild/moderate group. A direct severe vs. mild/moderate comparison could introduce additional complexity without necessarily clarifying the core biological distinctions we aimed to explore, especially given the more subtle differences expected between these two groups.

To maintain clarity and interpretability, we chose to focus on changes relative to untreated controls. This design ensures a consistent reference point across comparisons, enabling a more controlled interpretation of transcriptomic changes driven by disease severity while minimizing confounding effects from inter-treatment variability.

Comments 27: Where does the cutoff log2FC>0.58 stem from?

Response 27: Thank you for your question. The cutoff of log₂ fold change +/- > 0.58 was selected as it corresponds to a +/-1.5-fold change, which is commonly used in transcriptomic and proteomic analyses to define biologically meaningful differences in expression. This threshold was used in combination with an adjusted p-value < 0.05 to balance sensitivity with stringency in identifying differentially expressed genes.

Comments 28: Figure 4: panel A does not illustrate what is described in results and figure legend. Although interesting, the data shown in Figure 4B and C do not provide an explanation for the observed differential effects of SFEVsevere and SFEVmild. Other factors appear to be responsible for this differential effect. Experiments that assess these factors would be of great benefit for the impact of the work. At present, the data in Figure 4 represent a characterization of the respective EVs, but should be presented earlier in the manuscript.

Response 28: Thank you for this constructive comment. We have edited figure 4A to better reflect the data discussed in the results and figure legend. We agree that the data in Figure 4 primarily reflect characterisation of SFEV cargo (e.g., NGF content and protein markers) rather than a mechanistic explanation for their differential functional effects on chondrocytes. Our intent with this figure was to highlight differences in EV-associated cargo that may contribute to functional outcomes, and to generate hypotheses for future mechanistic studies.

We also appreciate the suggestion to relocate Figure 4 earlier in the manuscript. While we considered this, we chose to keep it in its current position to allow the reader to first understand the biological effects (Figures 2–3) before introducing mechanistic insights.

Comments 29: Line 435: It is unclear whether the SFEV lysates were prepared from EVs from patients with severe or mild/moderate OA.

Response 29: Thank you for your comment. SFEV lysates were prepared from the same cohort of 13 patients used for EV characterisation. This included 7 mild/moderate OA, 4 severe OA, and 2 patients who did not meet classification criteria based on OKS and EQ5D. For grouped comparisons, only the 11 classified patients (7 mild/moderate and 4 severe) were included in statistical analyses. We have clarified this in the revised Methods, Results and figure 4 legend to ensure consistency. Methods “SFEV lysates were prepared from EVs isolated from the 13 patient samples used for EV characterisation. These included 7 classified as mild/moderate OA, 4 as severe OA, and 2 who did not meet the criteria for either group. For group-based comparisons, only lysates from the 11 classified patients were included in the analysis.”, results “Lysates were prepared from EVs isolated from 13 OA patients (7 mild/moderate, 4 severe, and 2 unclassified). Group-based comparisons were conducted using the 11 patients with complete classification data.”, figure 4 legend “(A) Schematic overview of experimental design showing chondrocyte stimulation with SFEVs, collection of conditioned media for secreted protein analysis (ELISA and Luminex), and direct SFEV lysis for EV cargo profiling. EVs used for these analyses were isolated from 13 OA patients (7 mild/moderate, 4 severe, and 2 unclassified); group comparisons were performed using n = 11 classified patients.”

Comments 30: Discussion: Again, the role of synovial inflammation in OA is highlighted, but the association of this fact to the present study is unclear. “Severe” OA, as assessed via questionnaires is not necessarily related to synovial inflammation. This issue is a major concern with the present work. As such, no difference in IL6 between EVs from severe and mild/moderate OA was found. Thus, inflammatory factors might not be the relevant agents that explains the different impact.

Response 30: All patients recruited this study exhibited significant synovitis and therefore understanding inflammatory responses induced by SFEVs is relevant. To this end, our data supports a functional inflammatory response induced by SFEVs — particularly those from severe OA patients. We observed a significant increase in IL-6 protein secretion by chondrocytes treated with SFEVs from both groups, with the most pronounced effect in the severe OA group (Figure 4B). Although IL6 mRNA expression was not significantly altered, the protein-level response suggests post-transcriptional regulation or EV-driven cytokine release. Furthermore, IL-6 was the top-ranked upstream regulator in our IPA analysis of DEGs following severe SFEV treatment. Inflammatory responses were not limited to IL-6; SFEV-treated chondrocytes also upregulated CXCL5 and ICAM1 — both well-established inflammatory mediators. Collectively, these findings support our interpretation that SFEVs from severe OA patients promote an inflammatory chondrocyte phenotype, even in the absence of differential IL-6 content in EV cargo.

Comments 31: Line 539-542: This is unclear, as – in fact – the authors should have an EV-depleted SF control in their hands, as they ultracentrifuged the SF for 16h, which by definition us a method for EV depletion. The resulting supernatant would represent such a EV-depleted SF control.

Response 31: Thank you for your comment. While ultracentrifugation at 100,000 × g for 16 h is indeed used for EV depletion in certain biofluids, synovial fluid presents unique challenges due to its high viscosity and complex matrix composition — even after hyaluronidase treatment. In our hands, ultracentrifugation alone did not reliably deplete EVs from synovial fluid, as confirmed by NTA, and attempts to generate an EV-depleted supernatant through multiple rounds of ultracentrifugation were unsuccessful without extensive dilution or chemical modification.

Because such modifications would alter the composition of the fluid and introduce confounding factors, we chose to use untreated synovial fluid (containing EVs) for chondrocyte treatments. We now clarify this in the revised Discussion and figure legends.

“Although inclusion of an EV-depleted synovial fluid control would have strengthened this comparison, reliable depletion was not feasible due to the high viscosity of synovial fluid. Methods that could remove EVs (e.g. repeated ultracentrifugation or size exclusion) would have altered the composition of the fluid, necessitating additional control groups and complicating interpretation. Therefore, full synovial fluid was used to maintain comparability and physiological relevance.”

Reviewer 4 Report

Comments and Suggestions for Authors

The authors evaluated how extracellular vesicles (EVs) isolated from the synovial fluids (SFs) of patients with mild/moderate and severe osteoarthritis (OA) impact chondrocyte functions, in particular factors implicated in protein matrix remodeling/degradation, inflammation, and nociception. They report that compared SFEVs from mild/moderate OA patients, SFEVs from severe OA patients promote distinct transcriptomics responses in chondrocytes that may contribute to aggravating cartilage damage and enhanced pain nociception. 

Comments: The manuscript is well-written. However, the author could strengthen the Materials & Methods to facilitate understanding of the experimental design, particularly the incubation of chondrocytes with EVs and SFs from patients with mild/moderate and severe OA. 

Please increase the size of the figure fonts to facilitate reading. 

Authors must indicate the total number of recruited patients in section 2.1.

Line 101: EVs from the synovial fluids of 13 patients were analyzed. How many were from mild/moderate and severe OA patients? The information is essential, as Table 1 presents 11 OA patients (7 mild/moderate OA & 4 severe OA). Fetal calf serum (FCS) contains extracellular vesicles. Was FCS centrifuged at 100,000g to deplete the EV content before preparing the chondrocyte growth media?

Materials and Methods: Please provide more details on how primary chondrocytes were treated and processed before the extraction of bulk RNA. 

Fig. 2 and text (line 197): It is unclear if the SFEVs and SFs added to the primary chondrocyte cultures are from patients with mild/moderate and severe OA. The same comment also applies to chondrocytes (isolated from mild/moderate OA or severe OA?). Please clarify these points since the synovial environment, disease duration, and joint damage severity can impact the chondrocyte phenotype. It is a matter for discussion. Furthermore, make clear that SFs are SFEV-free SFs. 

Lines 196-199:  It is unclear whether the SFs added to the culture chondrocytes were free of EVs. The authors should clarify this point in the figure legends or the "Materials and Methods" section.

Fig. 3D, 3F, and 3G: The number of samples analyzed varies from 11 to 13. How many SFEV samples are from patients with mild/moderate and severe OA? 

Figs 3 & 4: Are the chondrocytes used in the experiments from patients with mild/moderate and severe OA? If not, discuss the limits of the study and potential bias. 

Various MMPs and proteins are EV encapsulated. I congratulate the authors for validating that SFEVs act as cargos for several proteins (BNDF, NGF, MMP1, MMM13) and that proteins encapsulated in SFEVs contribute minimally to the contents of proteins secreted but SFEV-treated chondrocytes (Supplementary Figure 1). It is unclear if protein contents are for SFs and SFEVs from mild/moderate OA or severe OA patients (Supplementary Figure 1). The data (n=7-13) suggest the authors used samples from mild/moderate OA and severe OA patients. 

I am a bit puzzled by the NFG data. The authors highlight a significant positive correlation between SFEV NGF concentration/EV particles and VAS pain severity scores (Figure 4E). However, the chondrocytes secrete less NGF when incubated SFEVs that cargo NGF (Supplementary Figure 1). The statement regarding the potential mechanistic link between SFEVs and patient-reported pain perception seems a little tenuous. 

Author Response

Comments 1: However, the author could strengthen the Materials & Methods to facilitate understanding of the experimental design, particularly the incubation of chondrocytes with EVs and SFs from patients with mild/moderate and severe OA. 

Response 1: Thank you for this helpful suggestion. We have revised the Materials & Methods section to more clearly describe the experimental design, including the source of the chondrocytes, the origin of the SFEVs and synovial fluid used for treatment (mild/moderate or severe OA), and how patient groups were stratified.

Comments 2: Please increase the size of the figure fonts to facilitate reading. 

Response 2: In the revised manuscript we have increased the labelling of axis and text in all figure panels.

Comments 3: Authors must indicate the total number of recruited patients in section 2.1.

Response 3: Thank you for this suggestion. We have now added the total number of recruited patients (n = 71) to Section 2.1 of the Methods to clarify the cohort size used for stratification and analysis.

Comments 4: Line 101: EVs from the synovial fluids of 13 patients were analyzed. How many were from mild/moderate and severe OA patients? The information is essential, as Table 1 presents 11 OA patients (7 mild/moderate OA & 4 severe OA). Fetal calf serum (FCS) contains extracellular vesicles. Was FCS centrifuged at 100,000g to deplete the EV content before preparing the chondrocyte growth media?

Response 4: Thank you for these important observations. The 13 patients included in EV characterisation (NTA, ExoView, protein cargo) consisted of 7 mild/moderate OA, 4 severe OA, and 2 patients who did not meet the criteria for either group. This breakdown has now been clarified in the revised Results and Methods sections. All group-based analyses were conducted using the 11 patients with complete classification.

Regarding FCS, we confirm that no FCS was used during the 24-hour chondrocyte treatment period. To avoid any confounding effects from FCS-derived extracellular vesicles, all treatments with SFEVs or synovial fluid were performed in serum-free media. We have now clarified this point in the revised Methods section. “Treatments were carried out in serum-free media for 24 h.”

Comments 5: Materials and Methods: Please provide more details on how primary chondrocytes were treated and processed before the extraction of bulk RNA. 

Response 5: Thank you for your comment, we have now provided more details on this. “Cartilage was obtained from macroscopically intact regions of the joint from individual patients undergoing total knee arthroplasty. As anatomical compartmentalisation was not recorded, samples likely reflect a mixture of femoral, tibial, or patellar articular cartilage. Fresh cartilage was dissected into 1-3mm3 pieces, and digested using 2mg/ml collagenase (Merck, C9891) in DMEM (Merck, D6429) for 7 hours at 37˚C. Chondrocytes were cultured in standard monolayer and used at early passage (P2) to minimise dedifferentiation. The same donor-derived chondrocyte population (BMI: 28.36) was used across all treatment conditions to minimise inter-donor variability. No data were collected on OKS, EQ5D, synovitis, or VAS for this donor to avoid introducing bias based on inflammatory status or clinical scores that could inadvertently influence downstream interpretations. Chondrocytes were treated with either pooled SFEVs or pooled synovial fluid from patients with mild/moderate or severe OA. Pooled samples were created by combining equal volumes from four classified donors per group (n = 4 mild/moderate, n = 4 severe). Treatments were carried out in serum-free media for 24 h.”

Comments 6: Fig. 2 and text (line 197): It is unclear if the SFEVs and SFs added to the primary chondrocyte cultures are from patients with mild/moderate and severe OA. The same comment also applies to chondrocytes (isolated from mild/moderate OA or severe OA?). Please clarify these points since the synovial environment, disease duration, and joint damage severity can impact the chondrocyte phenotype. It is a matter for discussion. Furthermore, make clear that SFs are SFEV-free SFs. 

Response 6: Thank you for this important comment. We confirm that no EV-depleted supernatants were used in this study. All synovial fluid used for chondrocyte treatment was untreated whole synovial fluid, and not processed through ultracentrifugation prior to use. Although we initially intended to include an EV-depleted synovial fluid control, this proved technically challenging due to the high viscosity of synovial fluid — even after hyaluronidase treatment. Multiple rounds of ultracentrifugation failed to reliably deplete EVs without extensive dilution of the sample. Importantly, any alternative depletion method (e.g. size exclusion chromatography, filtration, or chemical precipitation) would have altered the composition of the synovial fluid itself, introducing additional variables and requiring separate controls. For this reason, we elected to use full synovial fluid (with EVs intact) to preserve physiological relevance and comparability with the EV-only condition. Our original discussion of this has now been further expanded to address this “Although inclusion of an EV-depleted synovial fluid control would have strengthened this comparison, reliable depletion was not feasible due to the high viscosity of synovial fluid. Methods that could remove EVs (e.g. repeated ultracentrifugation or size exclusion) would have altered the composition of the fluid, necessitating additional control groups and complicating interpretation. Therefore, full synovial fluid was used to maintain comparability and physiological relevance.”

SFEVs and synovial fluid used for chondrocyte stimulation were derived from patients classified as mild/moderate or severe OA, with pooled samples from four donors per group used for each treatment.

Chondrocytes were isolated from a single OA donor undergoing total knee replacement. This donor had a BMI of 28.36 (overweight category), and cells were used at passage 2 to minimise dedifferentiation. No data were collected on OKS, EQ5D, synovitis, or VAS for this donor. This was intentional — the donor was selected prior to classification, to avoid introducing bias based on inflammatory status or clinical scores that could inadvertently influence downstream interpretations. The same chondrocyte population was used across all treatment groups to reduce variability. This has been clarified in the methods section “Cartilage was obtained from macroscopically intact regions of the joint from individual patients undergoing total knee arthroplasty. As anatomical compartmentalisation was not recorded, samples likely reflect a mixture of femoral, tibial, or patellar articular cartilage. Fresh cartilage was dissected into 1-3mm3 pieces, and digested using 2mg/ml collagenase (Merck, C9891) in DMEM (Merck, D6429) for 7 hours at 37˚C. Chondrocytes were cultured in standard monolayer and used at early passage (P2) to minimise dedifferentiation. The same donor-derived chondrocyte population (BMI: 28.36) was used across all treatment conditions to minimise inter-donor variability. No data were collected on OKS, EQ5D, synovitis, or VAS for this donor to avoid introducing bias based on inflammatory status or clinical scores that could inadvertently influence downstream interpretations. Chondrocytes were treated with either pooled SFEVs or pooled synovial fluid from patients with mild/moderate or severe OA. Pooled samples were created by combining equal volumes from four classified donors per group (n = 4 mild/moderate, n = 4 severe). Treatments were carried out in serum-free media for 24 h.”

Comments 7: Lines 196-199:  It is unclear whether the SFs added to the culture chondrocytes were free of EVs. The authors should clarify this point in the figure legends or the "Materials and Methods" section.

Response 7: Thank you for your comment. We confirm that the synovial fluid used for chondrocyte treatment was whole (not EV-depleted). Due to the high viscosity of synovial fluid and limited depletion efficiency, ultracentrifugation was not effective in removing EVs. As now described in the discussion, alternative depletion methods would have altered the composition of the fluid and introduced additional variables.

Comments 8: Fig. 3D, 3F, and 3G: The number of samples analyzed varies from 11 to 13. How many SFEV samples are from patients with mild/moderate and severe OA? 

Response 8: Thank you for this comment, we have revised the figure legends and manuscript to ensure n numbers are accurate.

Comments 9: Figs 3 & 4: Are the chondrocytes used in the experiments from patients with mild/moderate and severe OA? If not, discuss the limits of the study and potential bias. 

Response 9: Chondrocytes were isolated from a single OA donor undergoing total knee replacement. This donor had a BMI of 28.36 (overweight category), and cells were used at passage 2 to minimise dedifferentiation. No data were collected on OKS, EQ5D, synovitis, or VAS for this donor. This was intentional — the donor was selected prior to classification, to avoid introducing bias based on inflammatory status or clinical scores that could inadvertently influence downstream interpretations. The same chondrocyte population was used across all treatment groups to reduce variability. This has been clarified in the methods section “Cartilage was obtained from macroscopically intact regions of the joint from individual patients undergoing total knee arthroplasty. As anatomical compartmentalisation was not recorded, samples likely reflect a mixture of femoral, tibial, or patellar articular cartilage. Fresh cartilage was dissected into 1-3mm3 pieces, and digested using 2mg/ml collagenase (Merck, C9891) in DMEM (Merck, D6429) for 7 hours at 37˚C. Chondrocytes were cultured in standard monolayer and used at early passage (P2) to minimise dedifferentiation. The same donor-derived chondrocyte population (BMI: 28.36) was used across all treatment conditions to minimise inter-donor variability. No data were collected on OKS, EQ5D, synovitis, or VAS for this donor to avoid introducing bias based on inflammatory status or clinical scores that could inadvertently influence downstream interpretations. Chondrocytes were treated with either pooled SFEVs or pooled synovial fluid from patients with mild/moderate or severe OA. Pooled samples were created by combining equal volumes from four classified donors per group (n = 4 mild/moderate, n = 4 severe). Treatments were carried out in serum-free media for 24 h.” and discussion “We acknowledge that the use of chondrocytes from a single OA donor may influence the generalisability of the findings, as inter-donor variability in inflammatory responsiveness, matrix turnover, and cellular phenotype can shape how chondrocytes respond to external stimuli, including EVs. It is possible that chondrocytes derived from a donor with more advanced joint damage or a more inflammatory tissue microenvironment might have exhibited either heightened or blunted responses to SFEVs and synovial fluid. Similarly, variation in donor age, BMI, or metabolic status could influence sensitivity to vesicle-associated signals. However, our use of a single chondrocyte population minimised variability and ensured that observed differences in cellular response were attributable to the applied EVs and synovial fluid, rather than underlying differences in the chondrocyte source. Future studies incorporating multiple chondrocyte donors stratified by clinical or molecular phenotype will be important for validating the broader relevance of these findings.”

Comments 10: Various MMPs and proteins are EV encapsulated. I congratulate the authors for validating that SFEVs act as cargos for several proteins (BNDF, NGF, MMP1, MMM13) and that proteins encapsulated in SFEVs contribute minimally to the contents of proteins secreted but SFEV-treated chondrocytes (Supplementary Figure 1). It is unclear if protein contents are for SFs and SFEVs from mild/moderate OA or severe OA patients (Supplementary Figure 1). The data (n=7-13) suggest the authors used samples from mild/moderate OA and severe OA patients. 

Response 10: Thank you for your kind comment and for highlighting this point. We confirm that the protein content data presented in Supplementary Figure 1 were generated from SFEVs and synovial fluid derived from the 13 patients used in our EV characterisation cohort. These included 7 classified as mild/moderate OA, 4 as severe OA, and 2 patients who did not meet classification criteria. We have now clarified this information in the Supplementary Figure 1 legend. Concentrations of BDNF, IL-6, NGF, IL1B, MMP1, and MMP13 were quantified by Luminex or ELISA in raw hyaluronidase-treated synovial fluid (SF) and corresponding SFEV lysates isolated from the same donor samples. SFEVs were obtained via ultracentrifugation and lysed with 0.5% Triton X-100 prior to analysis. Each point represents an individual patient (n = 13 total; 7 mild/moderate OA, 4 severe OA, 2 unclassified). Across all proteins tested, concentrations were consistently higher in whole synovial fluid compared to isolated SFEVs, indicating that only a small proportion of total protein is vesicle-associated. Bars represent mean ± SEM.”

Comments 11: I am a bit puzzled by the NFG data. The authors highlight a significant positive correlation between SFEV NGF concentration/EV particles and VAS pain severity scores (Figure 4E). However, the chondrocytes secrete less NGF when incubated SFEVs that cargo NGF (Supplementary Figure 1). The statement regarding the potential mechanistic link between SFEVs and patient-reported pain perception seems a little tenuous. 

Response 11: Thank you for this comment. We would like to clarify a misunderstanding — Supplementary Figure 1 presents protein cargo levels of NGF measured in lysed synovial fluid and lysed SFEVs, not NGF secreted by chondrocytes. In contrast, Figure 4B shows NGF secretion by chondrocytes following treatment with SFEVs, and in that experiment, we observed a significant increase in NGF secretion in response to SFEVs from severe OA patients. We also note that NGF mRNA expression (Figure 4C) was significantly reduced with SFEV treatment, suggesting that the increase in protein secretion may occur through post-transcriptional regulation. This is not uncommon in cytokine biology and does not contradict the observed correlation between SFEV NGF content and patient-reported VAS pain scores (Figure 4E).

Round 2

Reviewer 3 Report

Comments and Suggestions for Authors

The authors have adequately responded to this reviewers remarks and questions. Thus, most concerns are now clarified. However, there is still one minor issues that should be acknowledged by the authors:

NTA is not an adequate method to quantify "EVs" (i.e., a pure population of EVs), as it basically detects all kind of particles. It is acceptable that the authors used NTA to estimate EV concentrations, but the wording should be unambiguous. See responses to e.g., comments 17, 23 and 31. Appropriate changes of wording should be made in the text of the manuscript at the respective positions.

Author Response

Comments 1: NTA is not an adequate method to quantify "EVs" (i.e., a pure population of EVs), as it basically detects all kind of particles. It is acceptable that the authors used NTA to estimate EV concentrations, but the wording should be unambiguous. See responses to e.g., comments 17, 23 and 31. Appropriate changes of wording should be made in the text of the manuscript at the respective positions. 

Response 1: We thank the reviewer for this clarification and have revised the manuscript text to ensure unambiguous wording. We now clearly state that NTA measures particle concentrations, not specific EV counts, and that other particles may be present in the preparation. The presence of EVs was confirmed using ExoView tetraspanin profiling. While we refer to these preparations as synovial fluid extracellular vesicles (SFEVs) throughout the manuscript, we acknowledge that these may include co-isolated non-EV particles. All references to “EV concentration” now reflect this more cautious interpretation.  

“As NTA detects particles based on size and Brownian motion, it may overestimate true EV concentration. Therefore, while we use the term ‘SFEVs’ to describe our preparations, these may contain a heterogeneous mix of particles. The presence of EVs was confirmed using tetraspanin profiling via ExoView.” 

Reviewer 4 Report

Comments and Suggestions for Authors

I have no further comments. 

Author Response

Reviewer comment 1: I have no further comments.

Response 1: Thank you for reviewing our revised article